# Bandits with Feedback Graphs and Switching Costs

**Raman Arora**
Dept. of Computer Science
Johns Hopkins University
Baltimore, MD 21204
arora@cs.jhu.edu

**Teodor V. Marinov**
Dept. of Computer Science
Johns Hopkins University
Baltimore, MD 21204
tmarino2@jhu.edu

**Mehryar Mohri**
Google Research &
Courant Institute of Math. Sciences
New York, NY 10012
mohri@google.com

## Abstract

We study the adversarial multi-armed bandit problem where the learner is supplied with partial observations modeled by a *feedback graph* and where shifting to a new action incurs a fixed *switching cost*. We give two new algorithms for this problem in the informed setting. Our best algorithm achieves a pseudo-regret of $\tilde{O}(\gamma(G)^{\frac{1}{3}} T^{\frac{2}{3}})$, where $\gamma(G)$ is the domination number of the feedback graph. This significantly improves upon the previous best result for the same problem, which was based on the independence number of $G$. We also present matching lower bounds for our result that we describe in detail. Finally, we give a new algorithm with improved policy regret bounds when partial counterfactual feedback is available.

## 1   Introduction

A general framework for sequential learning is that of online prediction with expert advice [Littlestone and Warmuth, 1994, Cesa-Bianchi et al., 1997, Freund and Schapire, 1997], which consists of repeated interactions between a learner and the environment. The learner maintains a distribution over a set of experts or actions. At each round, the loss assigned to each action is revealed. The learner incurs the expected value of these losses for their current distribution and next updates her distribution. The learner's goal is to minimize her *regret*, which, in the simplest case, is defined as the difference between the cumulative loss over a finite rounds of interactions and that of the best expert in hindsight.

The scenario just described corresponds to the so-called *full information* setting where the learner is informed of the loss of all actions at each round. In the *bandit setting*, only the loss of the action they select is known to the learner. These settings are both special instances of a general model of online learning with side information introduced by Mannor and Shamir [2011], where the information available to the learner is specified by a *feedback graph*. In an undirected feedback graph, each vertex represents an action and an edge between vertices $a$ and $a'$ indicates that the loss of action $a'$ is observed when action $a$ is selected and vice-versa. The bandit setting corresponds to a feedback graph reduced to only self-loops at each vertex, the full information setting to a fully connected graph. Online learning with feedback graphs has been further extensively analyzed by Alon et al. [2013, 2017] and several other authors [Alon et al., 2015, Kocák et al., 2014, Cohen et al., 2016, Yun et al., 2018, Cortes et al., 2018].

In many applications, the learner also incurs a cost when switching to a new action. Consider, for example, a commercial bank that issues various credit card products, many of which are similar, e.g., different branded cards with comparable fees and interest rates. At each round, the bank offers a specific product to a particular sub-population (e.g., customers at a store). The payoff observed for this action also reveals feedback for related cards and similar sub-populations. At the same time, offering a different product to a group incurs a switching cost in terms of designing a new marketing campaign. Another example of a problem with feedback graph and switching costs is a large company seeking to allocate and reallocate employees to different tasks so that the productivity

is maximized. Employees with similar skills, e.g., technical expertise, people skills, can be expected to perform as well as each other on the same task. Reassigning employees between tasks, however, is associated with a cost for retraining and readjustment time. We refer the reader to Appendix B for more motivating examples.

The focus of this paper is to understand the fundamental tradeoffs between exploration and exploitation in online learning with feedback graphs and switching costs, and to design learning algorithms with provably optimal guarantees. We consider the general case of a feedback graph $G$ with a set of vertices or actions $V$. In the expert setting with no switching cost, the min-max optimal regret is achieved by the weighted-majority or the Hedge algorithm [Littlestone and Warmuth, 1994, Freund and Schapire, 1997], which is in $\Theta(\sqrt{\log(|V|)T})$. In the bandit setting, the extension of these algorithms, EXP3 [Auer et al., 2002], achieves a regret of $O(\sqrt{|V|\log(|V|)T})$. The min-max optimal regret of $\Theta(\sqrt{|V|T})$ can be achieved by the INF algorithm [Audibert and Bubeck, 2009]. The $\sqrt{|V|}$-term in the bandit setting is inherently related to the additional exploration needed to observe the loss of all actions.

The scenario of online learning with side information modeled by feedback graphs, which interpolates between the full information and the bandit setting, was introduced by Mannor and Shamir [2011]. When the feedback graph $G$ is fixed over time and is undirected, a regret in the order of $O(\sqrt{\alpha(G)\log(|V|)T})$ can be achieved, with a lower bound of $\Omega(\sqrt{\alpha(G)T})$, where $\alpha(G)$ denotes the independence number of $G$. There has been a large body of work studying different settings of this problem with time-varying graphs $(G_t)_{t=1}^T$, in both the directed or undirected cases, and in both the so-called *informed setting*, where, at each round, the learner receives the graph before selecting an action, or the *uninformed setting* where it is only made available after the learner has selected an action and updated its distribution [Alon et al., 2013, Kocák et al., 2014, Alon et al., 2015, Cohen et al., 2016, Alon et al., 2017, Cortes et al., 2018].

For the expert setting augmented with switching costs, the min-max optimal regret remains in $\tilde{\Theta}(\sqrt{\log(|V|)T})$. However, classical algorithms such as the Hedge or Follow-the-Perturbed-Leader [Kalai and Vempala, 2005] no more achieve the optimal regret bound. Several algorithms designed by Kalai and Vempala [2005], Geulen et al. [2010], Gyorgy and Neu [2014] achieve this min-max optimal regret. In the setting of bandits with switching costs, the lower bound was carefully investigated by Cesa-Bianchi et al. [2013] and Dekel et al. [2014] and shown to be in $\tilde{\Omega}(|V|^{\frac{1}{3}}T^{\frac{2}{3}})$. This lower bound is asymptotically matched by mini-batching the EXP3 algorithm, as proposed by Arora et al. [2012].

The only work we are familiar with, which studies both bandits with switching costs and side information is that of Rangi and Franceschetti [2019]. The authors propose two algorithms for time-varying feedback graphs in the uninformed setting. When reduced to the fixed feedback graph setting, their regret bound becomes $\tilde{O}(\alpha(G)^{\frac{1}{3}}T^{\frac{2}{3}})$. We note that, in the informed setting with a fixed feedback graph, this bound can be achieved by applying the mini-batching technique of Arora et al. [2012] to the EXP3-SET algorithm of Alon et al. [2013].

Our main contributions are two-fold. First, we propose two algorithms for online learning in the informed setting with a fixed feedback graph $G$ and switching costs. Our best algorithm admits a pseudo-regret bound in $\tilde{O}(\gamma(G)^{\frac{1}{3}}T^{\frac{2}{3}})$, where $\gamma(G)$ is the domination number of $G$. We note that the domination number $\gamma(G)$ can be substantially smaller than the independence number $\alpha(G)$ and therefore that our algorithm significantly improves upon previous work by Rangi and Franceschetti [2019] in the informed setting. We also extend our results to achieve a policy regret bound in $\tilde{O}(\gamma(G)^{\frac{1}{3}}T^{\frac{2}{3}})$ when partial counterfactual feedback is available. The $\tilde{O}(\gamma(G)^{\frac{1}{3}}T^{\frac{2}{3}})$ regret bound in the switching costs setting might seem at odds with a lower bound stated by Rangi and Franceschetti [2019]. However, the lower bound of Rangi and Franceschetti [2019] can be shown to be technically inaccurate (see Appendix C). Our second main contribution is a lower bound in $\tilde{\Omega}(T^{\frac{2}{3}})$ for any non-complete feedback graph. We also extend this lower bound to $\tilde{\Omega}(\gamma(G)^{\frac{1}{3}}T^{\frac{2}{3}})$ for a class of feedback graphs that we will describe in detail. In Appendix I, we show a lower bound for the setting of evolving feedback graphs, matching the originally stated lower bound in [Rangi and Franceschetti, 2019].

The rest of this paper is organized as follows. In Section 2, we describe in detail the setup we analyze and introduce the relevant notation. In Section 3, we describe our main algorithms and results. We

further extend our algorithms and analysis to the setting of online learning in reactive environments (Section 4). In Section 5, we present and discuss in detail lower bounds for this problem.

## 2 Problem Setup and Notation

We study a repeated game between an adversary and a player over $T$ rounds. For any $n \in \mathbb{N}$, we denote by $[n]$ the set of integers $\{1, \ldots, n\}$. At each round $t \in [T]$, the player selects an action $a_t \in V$ and incurs a loss $\ell_t(a_t)$, as well as a cost of one if switching between distinct actions in consecutive rounds ($a_t \neq a_{t-1}$). For convenience, we define $a_0$ as an element not in $V$ so that the first action always incurs a switching cost. The regret $R_T$ of any sequence of actions $(a_t)_{t=1}^T$ is thus defined by $R_T = \max_{a \in V} \sum_{t=1}^T \ell_t(a_t) - \ell_t(a) + M$, where $M = \sum_{t=1}^T 1_{a_t \neq a_{t-1}}$ is the number of action switches in that sequence. We will assume an oblivious adversary, or, equivalently, that the sequence of losses for all actions is determined by the adversary before the start of the game. The performance of an algorithm $\mathcal{A}$ in this setting is measured by its *pseudo-regret* $R_T(\mathcal{A})$ defined by

$$ R_T(\mathcal{A}) = \max_{a \in V} \mathbb{E}\left[ \sum_{t=1}^T \left( \ell_t(a_t) + 1_{a_t \neq a_{t-1}} \right) - \ell_t(a) \right], $$

where the expectation is taken over the player's randomized choice of actions. The *regret* of $\mathcal{A}$ is defined as $\mathbb{E}[R_T]$, with the expectation outside of the maximum. In the following, we will abusively refer to $R_T(\mathcal{A})$ as the *regret* of $\mathcal{A}$, to shorten the terminology.

We also assume that the player has access to an undirected graph $G = (V, E)$, which determines which expert losses can be observed at each round. The vertex set $V$ is the set of experts (or actions) and the graph specifies that, if at round $t$ the player selects action $a_t$, then, the losses of all experts whose vertices are adjacent to that of $a_t$ can be observed: $\ell_t(a)$ for $a \in N(a_t)$, where $N(a_t)$ denotes the neighborhood of $a_t$ in $G$ defined for any $u \in V$ by: $N(u) = \{v : (u, v) \in E\}$. We will denote by $deg(u) = |N(u)|$ the degree of $u \in V$ in graph $G$. We assume that $G$ admits a self-loop at every vertex, which implies that the player can at least observe the loss of their own action (bandit information). In all our figures, self-loops are omitted for the sake of simplicity.

We assume that the feedback graph is available to the player at the beginning of the game (*informed setting*). The *independence number* of $G$ is the size of a *maximum independent set* in $G$ and is denoted by $\alpha(G)$. The *domination number* of $G$ is the size of a *minimum dominating set* and is denoted by $\gamma(G)$. The following inequality holds for all graphs $G$: $\gamma(G) \leq \alpha(G)$ [Bollobás and Cockayne, 1979, Goddard and Henning, 2013]. In general, $\gamma(G)$ can be substantially smaller than $\alpha(G)$, with $\gamma(G) = 1$ and $\alpha(G) = |V| - 1$ in some cases. We note that all our results can be straightforwardly extended to the case of directed graphs.

## 3 An Adaptive Mini-batch Algorithm

In this section, we describe an algorithm for online learning with switching costs, using adaptive mini-batches. All proofs of results are deferred to Appendix D.

The standard exploration versus exploitation dilemma in the bandit setting is further complicated in the presence of a feedback graph: if a poor action reveals the losses of all other actions, do we play the poor action? The lower bound construction of Mannor and Shamir [2011] suggests that we should not, since it might be better to just switch between the other actions.

Adding switching costs, however, modifies the price of exploration and the lower bound argument of Mannor and Shamir [2011] no longer holds. It is in fact possible to show that EXP3 and its graph feedback variants switch too often in the presence of two good actions, thereby incurring $\Omega(T)$ regret, due to the switching costs. One way to deal with the switching costs problem is to adapt the fixed mini-batch technique of Arora et al. [2012]. That technique, however, treats all actions equally while, in the presence of switching costs, actions that provide additional information are more valuable.

We deal with the issues just discussed by adopting the idea that the mini-batch sizes could depend both on how favorable an action is and how much information an action provides about good actions.

---
**Algorithm 1** Algorithm for star graphs
---
**Input:** Star graph $G(V, E)$, learning rates $(\eta_t)$, exploration rate $\beta \in [0, 1]$, maximum mini-batch $\tau$.
**Output:** Action sequence $(a_t)_{t=1}^T$.
1: $q_1 = \frac{1}{|V|}$.
2: **while** $\sum_t \lfloor \tau_t \rfloor \leq T$ **do**
3:     $p_t = (1 - \beta)q_t + \beta\delta(r)$           % $\delta(r)$ is the Dirac distribution on $r$
4:     Draw $a_t \sim p_t$, set $\tau_t = p_t(r)\tau$
5:     **if** $a_{t-1} \neq r$ and $a_t \neq r$ **then**
6:         Set $a_t = a_{t-1}$
7:     **end if**
8:     Play $a_t$ for the next $\lfloor \tau_t \rfloor$ iterations
9:     Set $\widehat{\ell}_t(i) = \sum_{j=t}^{t+\lfloor \tau_t \rfloor - 1} \mathbb{I}(a_t = r)\frac{\ell_j(i)}{p_t(r)}$
10:     For all $i \in V$, $q_{t+1}(i) = \frac{q_t(i)\exp(-\eta_t\widehat{\ell}_t(i))}{\sum_{j \in V} q_t(j)\exp(-\eta_t\widehat{\ell}_t(j))}$
11:     $t = t + 1$
12: **end while**
---

## 3.1   Algorithm for Star Graphs

We start by studying a simple feedback graph case in which one action is adjacent to all other actions with none of these other actions admitting other neighbors. For an example see Figure 1.

We call such graphs *star graphs* and we refer to the action adjacent to all other actions as the *revealing action*. The revealing action is denoted by $r$. Since only the revealing action can convey additional information about other actions, we will select our mini-batch size to be proportional to the quality of this action. Also, to prevent our algorithm from switching between two non-revealing actions too often, we will simply disallow that and allow switching only between the revealing action and a non-revealing action. Finally, we will disregard any feedback a non-revealing action provides us. This simplifies the analysis of the regret of our algorithm. The pseudocode of the algorithm is given in Algorithm 1.

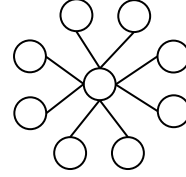

Figure 1: Example of a star graph.

The following intuition guides the design of our algorithm and its analysis. We need to visit the revealing action sufficiently often to derive information about all other actions, which is determined by the explicit exploration factor $\beta$. If $r$ is a good action, our regret will not be too large if we visit it often and spent a large amount of time in it. On the other hand if $r$ is poor, then the algorithm should not sample it often and, when it does, it should not spend too much time there. Disallowing the algorithm to directly switch between non-revealing actions also prevents it from switching between two good non-revealing actions too often. The only remaining question is: do we observe enough information about each action to be able to devise a low regret strategy? The following regret guarantee provides a precise positive response.

**Theorem 3.1.** *Suppose that the inequality $\mathbb{E}[\ell_t^2(i)] \leq \rho$ holds for all $t \leq T$ and all $i \in V$, for some $\rho$ and $\beta \geq \frac{1}{\tau}$. Then, for any action $a \in V$, Algorithm 1 admits the following guarantee:*

$$\mathbb{E}\left[\sum_{t=1}^T \ell_t(a_t) - \ell_t(a)\right] \leq \frac{\log(|V|)}{\eta} + T\eta\tau\rho + T\beta.$$

*Furthermore, the algorithm does not switch more than $2T/\tau$ times, in expectation.*

The exploration parameter $\beta$ is needed to ensure that $\tau_t = p_t(r)\tau \geq 1$, so that at every iteration of the while loop Algorithm 1 plays at least one action. The bound assumed on the second moment $\mathbb{E}[\ell_t^2(i)]$ might seem unusual since in the adversarial setting we do not assume a randomization of the losses. For now, the reader can just assume that this is a bound on the squared loss, that is, $\ell_t^2(i) \leq \rho$. The role of this expectation and the source of the randomness will become clear in Section 3.3. We note that the star graph admits independence number $\alpha(G) = |V| - 1$ and domination number $\gamma(G) = 1$.

In this case, the algorithms of Rangi and Franceschetti [2019] and variants of the mini-batching algorithm only guarantee a regret bound of the order $\tilde{O}(\alpha(G)^{\frac{1}{3}}T^{\frac{2}{3}})$, while Algorithm 1 guarantees a regret bound of the order $\tilde{O}(T^{\frac{2}{3}})$ when we set $\eta = 1/T^{\frac{2}{3}}$, $\tau = T^{\frac{2}{3}}$, and $\beta = 1/T^{\frac{1}{3}}$.

## 3.2 Algorithm for General Feedback Graphs

We now extend Algorithm 1 to handle arbitrary feedback graphs. The pseudocode of this more general algorithm is given in Algorithm 2.

---

**Algorithm 2** Algorithm for general feedback graphs

---

**Input:** Graph $G(V, E)$, learning rates $(\eta_t)$, exploration rate $\beta \in [0, 1]$, maximum mini-batch $\tau$.
**Output:** Action sequence $(a_t)_t$.
 1: Compute an approximate dominating set $R$
 2: $q_1 \equiv Unif(V), u \equiv Unif(R)$
 3: **while** $\sum_t \tau_t \leq T$ **do**
 4:     $p_t = (1 - \beta)q_t + \beta u$.
 5:     Draw $i \sim p_t$, set $\tau_t = p_t(r_i)\tau$, where $r_i$ is the dominating vertex for $i$ and set $a_t = i$.
 6:     **if** $a_{t-1} \notin R$ and $a_t \notin R$ **then**
 7:         Set $a_t = a_{t-1}$
 8:     **end if**
 9:     Play $a_t$ for the next $\lfloor \tau_t \rfloor$ iterations.
10:     Set $\widehat{\ell}_t(i) = \sum_{j=t}^{t+\lfloor \tau_t \rfloor - 1} \mathbb{I}(a_t = r_i) \frac{\ell_j(i)}{p_t(r_i)}$.
11:     For all $i \in V$, $q_{t+1}(i) = \frac{q_t(i)\exp(-\eta_t \widehat{\ell}_t(i))}{\sum_{j \in V} q_t(j)\exp(-\eta_t \widehat{\ell}_t(j))}$.
12:     $t = t + 1$.
13: **end while**

---

The first step of Algorithm 2 consists of computing an approximate minimum dominating set for $G$ using the Greedy Set Cover algorithm [Chvatal, 1979]. The Greedy Set Cover algorithm naturally partitions $G$ into disjoint star graphs with revealing actions/vertices in the dominating set $R$. Next, Algorithm 2 associates with each star-graph its revealing arm $r \in R$. The mini-batch size at time $t$ now depends on the probability $p_t(r)$ of sampling a revealing action $r$, as in Algorithm 1. There are several key differences, however, that we now point out. Unlike Algorithm 1, the mini-batch size can change between rounds even if the action remains fixed. This occurs when the newly sampled action is associated with a new revealing action in $R$, however, it is different from the revealing action. The above difference introduces some complications, because $\tau_t$ conditioned on all prior actions $a_{1:t-1}$ is still a random variable, while it is a deterministic in Algorithm 1. We also allow switches between any action and any vertex $r \in R$. This might seem to be a peculiar choice. For example, allowing only switches within each star-graph in the partition and only between revealing vertices seems more natural. Allowing switches between any vertex and any revealing action benefits exploration while still being sufficient for controlling the number of switches. If we further constrain the number of switches by using the more natural approach, it is possible that not enough information is received about each action, leading to worse regret guarantees. We leave the investigation of such more natural approaches to future work. Algorithm 2 admits the following regret bound.

**Theorem 3.2.** *For any $\beta \geq \frac{|R|}{\tau}$ The expected regret of Algorithm 2 is*

$$\frac{\log(|V|)}{\eta} + \eta\tau T + \beta T.$$

*Further, if the algorithm is augmented similar to Algorithm 7, then it will switch between actions at most $\frac{2T|R|}{\tau}$ times.*

Setting $\eta = 1/(|R|^{\frac{1}{3}}T^{\frac{2}{3}})$, $\tau = |R|^{\frac{2}{3}}T^{\frac{1}{3}}$ and $\beta = |R|^{\frac{1}{3}}/T^{\frac{1}{3}}$, recovers a pseudo-regret bound of $\tilde{O}(|R|^{\frac{1}{3}}T^{\frac{2}{3}})$, with an expected number of switches bounded by $2|R|^{\frac{1}{3}}T^{\frac{2}{3}}$. We note that $|R| = O(\gamma(G)\log(|V|))$ and thus the regret bound of our algorithm scales like $\gamma(G)^{\frac{1}{3}}$. Further, this is a strict improvement over the results of Rangi and Franceschetti [2019] as their result shows a scaling of $\alpha(G)^{\frac{1}{3}}$. The proof of Theorem 3.2 can be found in Appendix D.3.

---

**Algorithm 3** Corralling star-graph algorithms

---

**Input:** Feedback graph $G(V, E)$, learning rate $\eta$, mini-batch size $\tau$
**Output:** Action sequence $(a_t)_{t=1}^T$.
1: Compute an approximate minimum dominating set $R$ and initialize $|R|$ base star-graph algorithms, $B_1, B_2, \ldots, B_{|R|}$, with step size $\frac{\eta'}{2|R|}$, mini-batch size $\tau$ and exploration rate $1/\tau$ (Algorithm 1).
2: $T' = \frac{T}{\tau}, \beta = \frac{1}{T'}, \tilde{\beta} = \exp\left(\frac{1}{\log(T)}\right), \eta_{1,i} = \eta, \rho_{1,i} = 2|R|$ for all $i \in [|R|], q_1 = p_1 = \frac{1}{|R|}$
3: **for** $t = 1, \ldots, T'$ **do**
4:     Draw $i_t \sim p_t$
5:     **for** $j_t = (t-1)\tau + 1, \ldots, (t-1)\tau + \tau$ **do**
6:         Receive action $a_{j_t}^i$ from $B_i$ for all $i \in [|R|]$.
7:         Set $a_{j_t} = a_{j_t}^{i_t}$, play $a_{j_t}$ and observe loss $\ell_{j_t}(a_{j_t})$.
8:         Send $\frac{\ell_{j_t}(a_{j_t})}{p_t(i_t)}\mathbb{I}\{i = i_t\}$ as loss to algorithm $B_i$ for all $i \in [|R|]$.
9:         Update $\widehat{\ell}_t(i) = \widehat{\ell}_t(i) + \frac{1}{\tau}\frac{\ell_{j_t}(a_{j_t})}{p_t(i_t)}\mathbb{I}\{i = i_t\}$.
10:     **end for**
11:     Update $q_{t+1} =$ Algorithm 4$(q_t, \widehat{\ell}_t, \eta_t)$.
12:     Set $p_{t+1} = (1 - \beta)q_{t+1} + \beta\frac{1}{|R|}$.
13:     **for** $i = 1, \ldots, |R|$ **do**
14:         **if** $\frac{1}{p_t(i)} > \rho_{t,i}$ **then**
15:             Set $\rho_{t+1,i} = \frac{2}{p_t(i)}, \eta_{t+1,i} = \tilde{\beta}\eta_{t,i}$ and restart $i$-th star-graph algorithm, with updated step-size $\frac{\eta'}{\rho_{t+1,i}}$
16:         **else**
17:             Set $\rho_{t+1,i} = \rho_{t,i}, \eta_{t+1,i} = \eta_{t,i}$.
18:         **end if**
19:     **end for**
20: **end for**

---

## 3.3 Corralling Star Graph Algorithms

---

**Algorithm 4** Log-Barrier-OMD$(q_t, \ell_t, \eta_t)$

---

**Input:** Previous distribution $q_t$, loss vector $\ell_t$, learning rate vector $\eta_t$.
**Output:** Updated distribution $q_{t+1}$.
1: Find $\lambda \in [\min_i \ell_t(i), \max_i \ell_t(i)]$ such that $\sum_{i=1}^{|R|} \frac{1}{\frac{1}{q_t(i)} + \eta_{t,i}(\ell_t(i) - \lambda)} = 1$
2: Return $q_{t+1}$ such that $\frac{1}{q_{t+1}(i)} = \frac{1}{q_t(i)} + \eta_{t,i}(\ell_t(i) - \lambda)$.

---

An alternative natural method to tackle the general feedback graph problem is to use the recent corralling algorithm of Agarwal et al. [2016]. Corralling star graph algorithms was in fact our initial approach. In this section, we describe that technique, even though it does not seem to achieve an optimal rate. Here too, the first step consists of computing an approximate minimum dominating set. Next, we initialize an instance of Algorithm 1 for each star graph. Finally, we combine all of the star graph algorithms via a mini-batched version of the corralling algorithm of Agarwal et al. [2016]. Mini-batching is necessary to avoid switching between star graph algorithms too often. The pseudocode of this algorithm is given in Algorithm 3. Since during each mini-batch we sample a single star graph algorithm, we need to construct appropriate unbiased estimators of the losses $\ell_{j_t}$, which we feed back to the sampled star graph algorithm. The bound on the second moment of these estimators is exactly what Theorem 3.1 requires. Our algorithm admits the following guarantees.

**Theorem 3.3.** *Let $\tau = T^{\frac{1}{3}}/|R|^{\frac{1}{4}}, \eta = |R|^{\frac{1}{4}}/(40c\log(T')T^{\frac{1}{3}}\log(|V|))$, and $\eta' = 1/T^{\frac{2}{3}}$, where $c$ is a constant independent of $T, \tau, |V|$ and $|R|$. Then, for any $a \in V$, the following inequality holds for Algorithm 3:*

$$\mathbb{E}\left[\sum_{t=1}^T \ell_t(a_t) - \ell_t(a)\right] \leq \tilde{O}\left(\sqrt{|R|}\,T^{\frac{2}{3}}\right).$$

---

**Algorithm 5** Policy regret with side observations

---

**Input:** Feedback graph $G(V, E)$, learning rate $\eta$, mini-batch size $\tau$, where $\eta$ and $\tau$ are set as in Theorem 3.3.
**Output:** Action sequence $(a_t)_t$.
1: Transform feedback graph $G$ from $m$-tuples to actions and initialize Algorithm 2.
2: **for** $t = 1, \ldots, T/m$ **do**
3:     Sample action $a_t$ from $p_t$ generated by Algorithm 2 and play it for the next $m$ rounds.
4:     **if** $a_{t-1} = a_t$ **then**
5:         Observe mini-batched loss $\widehat{\ell}_t(a_t) = \frac{1}{m} \sum_{j=1}^m \ell_{(t-1)m+j}(a_t)$ and additional side observations. Feed mini-batched loss and additional side observations to Algorithm 2.
6:     **else**
7:         Set $\widehat{\ell}_t(a_t) = 0$ and set additional feedback losses to 0. Feed losses to Algorithm 2.
8:     **end if**
9: **end for**

---

*Furthermore, the expected number of switches of the algorithm is bounded by* $T^{\frac{2}{3}} |R|^{\frac{1}{3}}$.

This bound is suboptimal compared to the $\gamma(G)^{\frac{1}{3}}$-dependency achieved by Algorithm 2. We conjecture that this gap is an artifact of the analysis of the corralling algorithm of Agarwal et al. [2016]. However, we were unable to improve on the current regret bound by simply corralling.

## 4   Policy Regret with Partial Counterfactual Feedback

In this section, we consider games played against an *adaptive adversary*, who can select losses based on the player's past actions. In that scenario, the notion of pseudo-regret is no longer meaningful or interpretable, as pointed out by Arora et al. [2012]. Instead, the authors proposed the notion of *policy regret* defined by the following: $\max_{a \in V} \sum_{t=1}^T \ell_t(a_1, \ldots, a_t) - \sum_{t=1}^T \ell_t(a, \ldots, a)$, where the benchmark action $a$ does not depend on the player's actions. Since it is impossible to achieve $o(T)$ policy regret when the $t$-th loss is allowed to depend on all past actions of the player, the authors made the natural assumption that the adversary is $m$-memory bounded, that is that the $t$-th loss can only depend on the past $m$ actions chosen by the player. In that case, the known min-max policy regret bounds are in $\tilde{\Theta}(|V|^{\frac{1}{3}} T^{\frac{2}{3}})$ [Dekel et al., 2014], ignoring the dependency on $m$.

Here, we show that the dependency on $|V|$ can be improved in the presence of partial counterfactual feedback. We assume that partial feedback on losses with memory $m$ is available. We restrict the feedback graph to admitting only vertices for repeated $m$-tuples of actions in $V$, that is, we can only observe additional feedback for losses of the type $\ell_t(a, a, \ldots, a)$, where $a \in V$. For a motivating example, consider the problem of prescribing treatment plans to incoming patients with certain disorders. Two patients that are similar, for example patients in the same disease sub-type or with similar physiological attributes, when prescribed different treatments, reveal counterfactual feedback about alternative treatments for each other.

Our algorithm for incorporating such partial feedback to minimize policy regret is based on our algorithm for general feedback graphs (Algorithm 2). The learner receives feedback about $m$-memory bounded losses in the form of $m$-tuples. We simplify the representation by replacing each $m$-tuple vertex in the graph by a single action, that is vertex $(a, \ldots, a)$ represented as $a$.

As described in Algorithm 5, the input stream of $T$ losses is split into mini-batches of size $m$, indexed by $t$, such that $\widehat{\ell}_t(\cdot) = \frac{1}{m} \sum_{j=1}^m \ell_{(t-1)m+j}(\cdot)$. This sequence of losses, $(\widehat{\ell}_t)_{t=1}^{T/m}$, could be fed as input to Algorithm 2 if it were not for the constraint on the additional feedback. Suppose that between the $t$-th mini-batch and the $t + 1$-st mini-batch, Algorithm 2 decides to switch actions so that $a_{t+1} \neq a_t$. In that case, no additional feedback is available for $\widehat{\ell}_{t+1}(a_{t+1})$ and the algorithm cannot proceed as normal. To fix this minor issue, the feedback provided to Algorithm 2 is that the loss of action $a_{t+1}$ was 0 and all actions adjacent to $a_{t+1}$ also incurred 0 loss. This modification of losses cannot occur more than the number of switches performed by Algorithm 2. Since the expected number of switches is bounded by $O(\gamma(G)^{\frac{1}{3}} T^{\frac{2}{3}})$, the modification does not affect the total expected regret.

**Theorem 4.1.** *The expected policy regret of Algorithm 5 is bounded as* $\tilde{O}((m\gamma(G))^{\frac{1}{3}} T^{\frac{2}{3}})$.

The proof of the above theorem can be found in Appendix E. Let us point out that Algorithm 5 requires knowledge (or an upper bound) on the memory of the adversary, unlike the algorithm proposed by Arora et al. [2012]. We conjecture that this is due to the adaptive mini-batch technique of our algorithm. In particular, we believe that for $m$-memory bounded adversaries, it is necessary to repeat each sampled action $a_t$ at least $m$ times.

## 5 Lower Bound

The main tool for constructing lower bounds when switching costs are involved is the stochastic process constructed by Dekel et al. [2014]. The crux of the proof consists of a carefully designed multi-scale random walk. The two characteristics of this random walk are its depth and its width. At time $t$, the depth of the walk is the number of previous rounds on which the value of the current round depends. The width of the walk measures how far apart two rounds that depend on each other are in time. The loss of each action is equal to the value of the random walk at each time step, and the loss of the best action is slightly better by a small positive constant. The depth of the process controls how well the losses concentrate in the interval $[0,1]$[1]. The width of the walk controls the variance between losses of different actions and ensures it is impossible to gain information about the best action, unless one switches between different actions.

### 5.1 Lower Bound for Non-complete Graphs

We first verify that the dependence on the time horizon cannot be improved from $T^{\frac{2}{3}}$ for any feedback graph in which there is at least one edge missing, that is, in which there exist two vertices that do not reveal information about each other. Without loss of generality, assume that the two vertices not joined by an edge are $v_1$ and $v_2$. Take any vertex that is a shared neighbor and denote this vertex by $v_3$ (see Figure 2

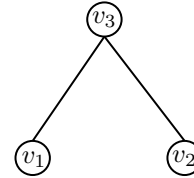

Figure 2: Feedback graph for switching costs

for an example). We set the loss for action $v_3$ and all other vertices to be equal to one. We now focus the discussion on the subgraph with vertices $\{v_1, v_2, v_3\}$. The losses of actions $v_1$ and $v_2$ are set according to the construction in [Dekel et al., 2014]. Since $\{v_1, v_2\}$ forms an independent set, the player would need to switch between these vertices to gain information about the best action. This is also what the lower bound proof of Rangi and Franceschetti [2019] is based upon. However, it is important to realize that the construction in Dekel et al. [2014] also allows for gaining information about the best action if its loss is revealed together with some other loss constructed from the stochastic process. In that case, playing vertex $v_3$ would provide such information. This is a key property which Rangi and Franceschetti [2019] seem to have missed in their lower bound proof. We discuss this mistake carefully in Appendix C and provide a lower bound matching what the authors claim in the *uninformed* setting in Appendix I. Our discussion suggests that we should set the price for revealing information about multiple actions according to the switching cost and this is why the losses of all vertices outside of the independent set are equal to one. We note that the losses of the best action are much smaller than one sufficiently often, so that enough instantaneous regret is incurred when pulling action $v_3$. Our main result follows and its proof can be found in Appendix F.

**Theorem 5.1.** *For any non-complete feedback graph $G$, there exists a sequence of losses on which any algorithm $\mathcal{A}$ in the informed setting incurs expected regret at least*

$$R_T(\mathcal{A}) \geq \Omega\left(\frac{T^{\frac{2}{3}}}{\log(T)}\right).$$

### 5.2 Lower Bound for Disjoint Union of Star Graphs

How do we construct a lower bound for a disjoint union of star graphs? First, note that if two adjacent vertices are allowed to admit losses set according to the stochastic process and one of them is the best vertex, then we could distinguish it in time $O(\sqrt{T})$ by repeatedly playing the other vertex. This suggests that losses set according to the stochastic process should be reserved for vertices in an

independent set. Second, it is important to keep track of the amount of information revealed by common neighbors.

Consider the feedback graph of Figure 3. This disjoint union of star graphs admits a domination number equal to four and its minimum dominating set is denoted by $\{v_1, v_2, v_3, v_4\}$. Probably the most natural way to set up the losses of the vertices is to set the losses of the maximum independent set, which consists of the colored vertices, according to the construction of Dekel et al. [2014] and the losses of the minimum dominating set

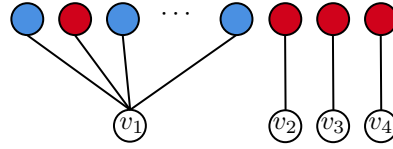

Figure 3: Disjoint union of star graphs.

equal to one. Let $v_1$ be the vertex with highest degree. Any time the best action is sampled to be not adjacent to $v_1$, switching between that action and $v_1$ reveals $deg(v_1)$ information about it. On the other hand, no matter how we sample the best action as a neighbor of $v_1$, it is then enough to play $v_1$ to gain enough information about it. If $I$ denotes the maximum independent set, the above reasoning shows that only $O(T^{\frac{2}{3}}|I|/deg(v_1))$ rounds of switching are needed to distinguish the best action. Since $deg(v_1)$ can be made arbitrarily large and thus $|I|/deg(v_1)$ gets arbitrary close to one, we see that the regret lower bound becomes independent of the domination number and equal to $\tilde{\Omega}(T^{\frac{2}{3}})$.

We now present a construction for the disjoint union of star graphs which guarantees a lower bound of the $\tilde{\Omega}(\gamma(G)^{\frac{1}{3}}T^{\frac{2}{3}})$. The idea behind our construction is to choose an independent set such that none of its members have a common neighbor, thereby avoiding the problem described above. We note that such an independent set cannot have size greater than $\gamma(G)$. Let $R$ be the set of revealing vertices for the star graphs. We denote by $V_i$ the set of vertices associated with the star graph with revealing vertex $v_i$. To construct the losses, we first sample an *active* vertex for each star graph from its leaves. The active vertices are represented in red in Figure 3. This forms an independent set $I$ indexed by $R$. Next, we follow the construction of Dekel et al. [2014] for the vertices in $I$, by first sampling a best vertex uniformly at random from $I$ and then setting the losses in $I$ according to the multi-scale random walk. All other losses are set to one. For any star graph consisting of a single vertex, we treat the vertex as a non-revealing vertex. This construction guarantees the following.

**Theorem 5.2.** *The expected regret of any algorithm $\mathcal{A}$ on a disjoint union of star graphs is lower bounded as follows:*

$$R_T(\mathcal{A}) \geq \Omega\left(\frac{\gamma(G)^{\frac{1}{3}}T^{\frac{2}{3}}}{\log(T)}\right).$$

The proof of this theorem can be found in Appendix G. This result can be viewed as a consequence of that of Dekel et al. [2014] but it can also be proven in alternative fashion. The general idea is to count the amount of information gained for the randomly sampled best vertex. For example, a strategy that switches between two revealing vertices $v_i$ and $v_j$ will gain information proportional to $deg(v_i)deg(v_j)$. The lower bound follows from carefully counting the information gain of switching between revealing vertices. This counting argument can be generalized beyond the disjoint union of star graphs, by considering an appropriate pair of minimal dominating/maximal independent sets. We give an argument for the disjoint union of star graphs in Appendix G and leave a detailed argument for general graphs to future work.

## 6 Conclusion

We presented an extensive analysis of online learning with feedback graphs and switching costs in the adversarial setting, a scenario relevant to several applications in practice. We gave a new algorithm whose regret guarantee only depends on the domination number. We also presented a matching lower bound for a family of graphs that includes disjoint unions of star graphs. The technical tools introduced in our proofs are likely to help derive a lower bound for all graph families. We further derived an algorithm with more favorable policy regret guarantees in the presence of feedback graphs.

## Acknowledgements

This research was partly supported by NSF BIGDATA grants IIS-1546482 and NSF IIS-1838139, and by NSF CCF-1535987, NSF IIS-1618662, and a Google Research Award.

## Footnotes

[1]Technically, the losses are always clipped between $[0,1]$.

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
