[Supplementary Material]

# Bandits with Feedback Graphs and Switching Costs

**Raman Arora**
Dept. of Computer Science
Johns Hopkins University
Baltimore, MD 21204
arora@cs.jhu.edu

**Teodor V. Marinov**
Dept. of Computer Science
Johns Hopkins University
Baltimore, MD 21204
tmarino2@jhu.edu

**Mehryar Mohri**
Google Research &
Courant Institute of Math. Sciences
New York, NY 10012
mohri@google.com

## Abstract

We study the adversarial multi-armed bandit problem where the learner is supplied with partial observations modeled by a *feedback graph* and where shifting to a new action incurs a fixed *switching cost*. We give two new algorithms for this problem in the informed setting. Our best algorithm achieves a pseudo-regret of $\tilde{O}(\gamma(G)^{\frac{1}{3}} T^{\frac{2}{3}})$, where $\gamma(G)$ is the domination number of the feedback graph. This significantly improves upon the previous best result for the same problem, which was based on the independence number of $G$. We also present matching lower bounds for our result that we describe in detail. Finally, we give a new algorithm with improved policy regret bounds when partial counterfactual feedback is available.

## 1 Introduction

A general framework for sequential learning is that of online prediction with expert advice [Littlestone and Warmuth, 1994, Cesa-Bianchi et al., 1997, Freund and Schapire, 1997], which consists of repeated interactions between a learner and the environment. The learner maintains a distribution over a set of experts or actions. At each round, the loss assigned to each action is revealed. The learner incurs the expected value of these losses for their current distribution and next updates her distribution. The learner's goal is to minimize her *regret*, which, in the simplest case, is defined as the difference between the cumulative loss over a finite rounds of interactions and that of the best expert in hindsight.

The scenario just described corresponds to the so-called *full information* setting where the learner is informed of the loss of all actions at each round. In the *bandit setting*, only the loss of the action they select is known to the learner. These settings are both special instances of a general model of online learning with side information introduced by Mannor and Shamir [2011], where the information available to the learner is specified by a *feedback graph*. In an undirected feedback graph, each vertex represents an action and an edge between vertices $a$ and $a'$ indicates that the loss of action $a'$ is observed when action $a$ is selected and vice-versa. The bandit setting corresponds to a feedback graph reduced to only self-loops at each vertex, the full information setting to a fully connected graph. Online learning with feedback graphs has been further extensively analyzed by Alon et al. [2013, 2017] and several other authors [Alon et al., 2015, Kocák et al., 2014, Cohen et al., 2016, Yun et al., 2018, Cortes et al., 2018].

In many applications, the learner also incurs a cost when switching to a new action. As an example, the learner may be a stock market investor who is charged a fixed commission when selling one stock or buying another (switching cost), but who may be exempt from additional fees when keeping their position in a stock. Similarly, an investor can sign a contract with an expert giving market advice, which, if broken, entails a termination fee. We assume that each expert works for a parent company and each parent company is willing to share the predictions made by its experts, together with the incurred losses. Another example of a problem with

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

# A  Related Work

We now discuss the work involving online learning with feedback graphs carefully. Most of the work we discuss deals with a feedback graph sequence $(G_t)_{t=1}^{T}$. The work of Mannor and Shamir [2011] is the first to study the online learning problem when feedback graphs model which losses the player gets to observe after choosing an action. Their work proposes two algorithms, the ExpBan, which has regret $O(\sqrt{\sum_{t=1}^{T} \bar{\chi}(G_t)})$, where $\bar{\chi}(G)$ is the clique partition number, and the ELP algorithm which has regret $O(\sqrt{\sum_{t=1}^{T} \alpha(G_t)})$. They also show a regret lower bound when $G_t = G$ for all $G$ of the order $\Omega(\sqrt{\alpha(G)T})$. The work of Alon et al. [2013] improves on that Mannor and Shamir [2011] in two significant ways. First the authors consider a setting in which the feedback graphs are directed and can be observed only after taking an action. Secondly the provided algorithms even for the informed setting are more efficient than the ones in Mannor and Shamir [2011]. Their algorithm Exp3-SET has regret $\tilde{O}(\sqrt{\sum_{t=1}^{T} \mathbf{mas}(G_t)})$ for the uninformed setting with directed feedback graphs. Here $\mathbf{mas}(G_t)$ is the size of the maximum acyclic subgraph of $G_t$. When considering the undirected setting $\mathbf{mas}(G_t)$ can be replaced by $\alpha(G_t)$. In the informed setting Alon et al. [2013] propose the algorithm Exp3-DOM, which requires approximating or computing a minimum dominating set of $G_t$. Kocák et al. [2014] avoid such tedious computation with their algorithm Exp3-IX. The regret achieved by their algorithm is of the order $\tilde{O}(\sqrt{\sum_{t=1}^{T} \alpha(G_t)})$ even in the uninformed setting. The paper also extends the implicit exploration trick used by Exp3-IX to Follow the Perturbed Leader and solves the combinatorial bandit problem with side observations, where at each round the player is permitted to select $m$ out of the $|V|$ available actions. The achieved regret is of the order $\tilde{O}(m^{2/3}\sqrt{\sum_{t=1}^{T} \alpha(G_t)})$. In Alon et al. [2015] the authors consider a setting where the feedback graph system is fixed i.e. $G_t = G$ for all $t \in [T]$, however, the graph need not have self loops. The authors distinguish between three settings. First a setting in which each vertex either has a self loop or is revealed by all other vertices, called the strongly observable setting. The second setting assumes that every vertex is revealed by some other vertex but there exists at least one vertex which is not strongly observable. This setting is called the weakly observable setting. The third setting is that of some vertex not being revealed by any other vertex. This is called the not observable setting. Alon et al. [2015] show that the regret bounds are respectively $\tilde{\Theta}(\sqrt{\alpha(G)T})$ in the strongly observable setting, $\tilde{\Theta}(\gamma(G)^{1/3}T^{2/3})$ in the weakly observable setting and $\Theta(T)$ in the not observable setting. The work of Cohen et al. [2016] studies a setting where the feedback graph is never fully revealed to the player. They show that if the feedback graph and the losses are generated by the adversary a lower bound for the regret of any strategy is $\Omega(\sqrt{|V|T})$, which matches the lower bound of the bandit setting. In contrast it is possible to recover a $\tilde{\Theta}(\sqrt{\alpha(G)T})$ regret bound if the losses are stochastic.

We also note that online learning with feedback graphs has also been studied in the setting of stochastic losses by numerous works [Caron et al., 2012, Buccapatnam et al., 2014, Wu et al., 2015a,b, Tossou et al., 2017, Liu et al., 2018], however, we chose not to discuss these works here as our focus is on the adversarial case.

# B  Motivating Examples

In this section we provide more motivating examples for studying the problem of online learning with feedback graphs and switching costs.

**Credit card products:**  Consider a commercial bank that issues various credit card products, many of which are similar, e.g., different branded cards with comparable fees and interest rates. At each round, the bank offers a specific product to a particular sub-population (e.g., customers at a store). The payoff observed for this action also reveals feedback for related cards and similar sub-populations. At the same time, offering a different product to a group incurs a switching cost in terms of designing a new marketing campaign.

**Store location:**  In certain states grocery stores are allowed to sell liquor, however, if a store brand has more than seven stores in a city, only seven of the stores are allowed to sell liquor. Switching

which store is selling liquor comes at a cost since a new liquor licence is required. If two stores in different cities have similar customer demographic, they reveal information about each other's sales and can be predictive of liquor sales revenue.

**Transportation logistics:** Logistic companies within the European Union are tasked with efficiently moving cargo within and between countries. Cargo is usually moved through ground transportation in the form of freight trucks. Since most logistic companies do not own their own trucks or drivers they have to chose among a set of truck companies serving different routes. A Truck companies like loyal customers and prefer working with logistic companies which regularly use the truck company's service. If a logistic company, however, decides to switch between several truck companies among serving the same route along a short period of time, the truck companies will raise their prices or altogether decide to not take more orders from the logistic company. Additional information is in the following form. A truck company offers a service between Paris and Berlin. The same truck company offers a service between Barcelona and Amsterdam. Since the routes are similar, the logistic company expects the utility of using this truck company for one of the routes to be close to the utility of using it for the other route.

**Moving between houses:** Students usually rent houses throughout their undergraduate and graduate studies. Moving between houses is a costly and time consuming process. However, students also get additional information when choosing a property to move to based on their current landlord/managing company. For example if a landlord is good but the property has some problems, the students might want to move to a different property managed by the same landlord.

**Oil field drilling:** A company in the oil business wants to decide where to setup drilling sites. They only have partial information about areas and can assume that two areas within some range have equal likelihood to have the same amount of oil resources. Switching between possible drilling sites is costly as it requires developing infrastructure. Further, switching between already build oil rigs and refineries requires relocating personal which is costly.

## C  Lower Bound of Rangi and Franceschetti [2019]

While going through the proof of Theorem 1 in Rangi and Franceschetti [2019], we came across an important technical mistake. In page 2 of the supplementary material, in the paragraph after Equation 8, the authors state that, at a single time instance, the loss of only one single action can be observed from the independent set in their construction. This is not correct since a player's strategy can play an action that is not in the independent set but is adjacent to two or more vertices in the independent set.

The problem with this statement becomes apparent when one considers a fixed feedback graph system, i.e., $G_t = G, \forall t \in [T]$, where $G$ is a star graph. In that case, the construction of the losses by Rangi and Franceschetti [2019] amounts to sampling a best action from the leaves of $G$, setting its loss to be $\epsilon_1$ smaller than the loss of all other actions in the leaves of $G$, and setting the revealing action to be $\epsilon_2$ larger than the losses in the leaves of $G$. The losses of the remaining actions are set according to the stochastic process of Dekel et al. [2014]. With these choice of losses and $\epsilon_1$ and $\epsilon_2$ set according to what the authors suggest, a very simple strategy is information-theoretically optimal: the player only needs to play the revealing action $T^{2/3}$ times to distinguish which of the leaves of $G$ contains the best action. This strategy would actually incur expected regret of the order $\tilde{\Theta}(\sqrt{T})$.

Let $\alpha(G_{1:T})$ denote the largest cardinality among all intersections of independent sets of the sequence $(G_t)_{t=1}^T$. A lower bound of $\tilde{\Omega}(\alpha(G_{1:T})^{1/3}T^{2/3})$ is still possible under additional assumptions about how the feedback graph system is generated in the *uninformed* setting. In particular, we show that if we allow the feedback graphs to be chosen by the adversary, there still exists a sequence of feedback graphs for which the lower bound is $\tilde{\Omega}(\alpha(G_{1:T})^{1/3}T^{2/3})$, while for each $G_t$, we have $\gamma(G_t) = 1$. This construction is presented in Section I with the main result stated in Theorem I.3.

## D  Proofs from Section 3

### D.1  Approximation to Minimum Dominating Set

---

**Algorithm 6** Greedy algorithm for minimum dominating set

---

**Input:** An undirected graph $G(V, E)$
**Output:** A dominating set $S$
 1: $R = \emptyset$
 2: **if** $V == \emptyset$ **then**
 3:     Return $S$
 4: **else**
 5:     Find $v \in V$ s.t. $deg(v)$ is maximized
 6:     $R = S \bigcup \{v\}$
 7:     $V = V \setminus \{\{v\} \bigcup N(v)\}$ and update $G$ to be the induced graph on the new set of vertices $V$.
 8: **end if**

---

The following notes http://ac.informatik.uni-freiburg.de/teaching/ss_12/netalg/lectures/chapter7.pdf provide us with a proof that the greedy Algorithm 6 returns a dominating set $R$ which is $2 + \log(\Delta)$ approximation to the smallest size minimal dominating set, where $\Delta$ is the maximum degree if $G$. It is possible to implement the algorithm so that it has total runtime of the order $O((|V| + |E|) \log(V))$ (e.g. http://homepage.cs.uiowa.edu/~sriram/3330/spring17/greedyMDS.pdf). We note that this is essentially the Greedy Set Cover algorithm of Chvatal [1979] and that it is possible to extend to directed graphs, by replacing the degree of $v$ by the out-degree of $v$ and the neighbours of $v$ by just the vertices which have in-going edge from $v$.

### D.1.1 Adaptive Mini-batching for Star Graphs

The proof of Theorem 3.1 begins by considering a slightly modified version of Algorithm 1. In particular we remove lines 5 through 7 which disallow switching between non-revealing actions. This intuitively should not change the policy which Algorithm 1 produces as such switches do not provide any new information to the algorithm. For convenience of the reader we give the pseudo-code of the modified algorithm in Algorithm 7, where the lines in red are commented out and are not part of the algorithm.

---

**Algorithm 7** Algorithm for star graphs (modified)

---

**Input:** Star graph $G(V, E)$, learning rate sequence $(\eta_t)$, exploration rate $\beta \in [0, 1]$, maximum mini-batch $\tau$.
**Output:** Action sequence $(a_t)_t$.
 1: $q_1 \equiv Unif(V)$.
 2: **while** $\sum_t \tau_t \leq T$ **do**
 3:     $p_t = (1 - \beta)q_t + \beta\delta(r)$.
 4:     Draw $a_t \sim p_t$, set $\tau_t = p_t(r)\tau$.
 5:     **if** $a_{t-1} \neq r$ and $a_t \neq r$ **then**
 6:         Set $a_t = a_{t-1}$
 7:     **end if**
 8:     Play $a_t$ for the next $\lfloor \tau_t \rfloor$ iterations.
 9:     Set

$$\widehat{\ell}_t(i) = \sum_{j=t}^{t+\lfloor \tau_t \rfloor - 1} \mathbb{I}(a_t = r)\frac{\ell_j(i)}{p_t(r)}.$$

10:     For all $i \in V$, $q_{t+1}(i) = \frac{q_t(i)\exp(-\eta_t \widehat{\ell}_t(i))}{\sum_{j \in V} q_t(j)\exp(-\eta_t \widehat{\ell}_t(j))}$.
11:     $t = t + 1$.
12: **end while**

---

Algorithm 7 comes with the following regret guarantee.

**Theorem D.1.** *Suppose that for all $t \leq T$ and all $i \in V$ it holds that $\mathbb{E}[\ell_t(i)^2] \leq \rho$ and $\beta \geq \frac{1}{\tau}$. Then Algorithm 7 produces an action sequence $(a_t)_{t=1}^T$ satisfying:*

$$\mathbb{E}\left[\sum_{t=1}^T \ell_t(a_t) - \ell_t(a)\right] \leq \frac{\log(|V|)}{\eta} + T\eta\tau\rho + T\beta,$$

*for any $a \in V$.*

*Proof.* Since $\beta \geq \frac{1}{\tau}$, this implies that $\lfloor \tau_t \rfloor \geq 1$ and the algorithm terminates, producing an action sequence $(a_t)_{t=1}^T$. Let $i_t^*$ be the best action at time $t$ and let $L_{t,*} = \sum_{s=1}^t \ell_s(i_t^*)$. Let $w_t(i) = \exp\left(-\eta \sum_{j=1}^{t-1} \widehat{\ell}_j(i)\right)$ and $W_t = \sum_{i \in V} w_t(i)$. We have

$$\log\left(\frac{W_{t+1}}{w_{t+1}(i_{t+1}^*)}\right) - \log\left(\frac{W_t}{w_t(i_t^*)}\right) = \eta(L_{t+1,*} - L_{t,*})$$

$$+ \log\left(\frac{\sum_{i \in V} w_t(i) \exp\left(-\eta \sum_{j=t}^{t+\lfloor \tau_t \rfloor - 1} \mathbb{I}(a_t = r)\frac{\ell_j(i)}{p_t(r)}\right)}{W_t}\right)$$

$$= \eta(L_{t+1,*} - L_{t,*})$$

$$+ \log\left(\sum_{i \in V} q_t(i) \exp\left(-\eta \sum_{j=t}^{t+\lfloor \tau_t \rfloor - 1} \mathbb{I}(a_t = r)\frac{\ell_j(i)}{p_t(r)}\right)\right)$$

$$\leq \eta(L_{t+1,*} - L_{t,*}) - 1$$

$$+ \sum_{i \in V} q_t(i) \exp\left(-\eta \sum_{j=t}^{t+\lfloor \tau_t \rfloor - 1} \mathbb{I}(a_t = r)\frac{\ell_j(i)}{p_t(r)}\right)$$

$$\leq \eta(L_{t+1,*} - L_{t,*}) - \eta\frac{\mathbb{I}(a_t = r)}{p_t(r)}\sum_{i \in V} q_t(i) \sum_{j=t}^{t+\lfloor \tau_t \rfloor - 1} \ell_j(i)$$

$$+ \frac{\eta^2}{2}\frac{\mathbb{I}(a_t = r)}{p_t(r)^2}\sum_{i \in V} q_t(i)\left(\sum_{j=t}^{t+\tau_t - 1} \ell_j(i)\right)^2,$$

where the first inequality follows from $\log(x) \leq x - 1$ for all $x > 0$ and the second inequality follows from $e^{-x} \leq 1 - x + x^2/2$ for $x \geq 0$. Rearranging terms in the above and taking expectation

we have

$$\mathbb{E}\left[\mathbb{E}\left[\frac{\mathbb{I}(a_t=r)}{p_t(r)}\sum_{i\in V}q_t(i)\sum_{j=t}^{t+\lfloor\tau_t\rfloor-1}\ell_j(i)|a_{1:t-1}\right]\right]\le\frac{1}{\eta}\mathbb{E}\left[\log\left(\frac{W_t}{w_t(i_t^*)}\right)-\log\left(\frac{W_{t+1}}{w_{t+1}(i_{t+1}^*)}\right)\right]$$

$$+\frac{\eta}{2}\mathbb{E}\left[\mathbb{E}\left[\frac{\mathbb{I}(a_t=r)}{p_t(r)^2}\sum_{i\in V}q_t(i)\left(\sum_{j=t}^{t+\tau_t-1}\ell_j(i)\right)^2|a_{1:t-1}\right]\right]+\mathbb{E}[L_{t+1,*}-L_{t,*}]$$

$$\implies$$

$$\mathbb{E}\left[\sum_{i\in V}q_t(i)\sum_{j=t}^{t+\lfloor\tau_t\rfloor-1}\ell_j(i)\right]\le\frac{1}{\eta}\mathbb{E}\left[\log\left(\frac{W_t}{w_t(i_t^*)}\right)-\log\left(\frac{W_{t+1}}{w_{t+1}(i_{t+1}^*)}\right)\right]$$

$$+\frac{\eta}{2}\mathbb{E}\left[\frac{1}{p_t(r)}\sum_{i\in V}q_t(i)\left(\sum_{j=t}^{t+\tau_t-1}\ell_j(i)\right)^2\right]+\mathbb{E}[L_{t+1,*}-L_{t,*}]$$

$$\implies$$

$$\mathbb{E}\left[\sum_{i\in V}q_t(i)\sum_{j=t}^{t+\lfloor\tau_t\rfloor-1}\ell_j(i)\right]\le\frac{1}{\eta}\mathbb{E}\left[\log\left(\frac{W_t}{w_t(i_t^*)}\right)-\log\left(\frac{W_{t+1}}{w_{t+1}(i_{t+1}^*)}\right)\right]$$

$$+\frac{\eta}{2}\mathbb{E}\left[\frac{1}{p_t(r)}\sum_{i\in V}q_t(i)\tau_t\sum_{j=t}^{t+\tau_t-1}\ell_j(i)^2\right]+\mathbb{E}[L_{t+1,*}-L_{t,*}]$$

$$\implies$$

$$\mathbb{E}\left[\sum_{i\in V}q_t(i)\sum_{j=t}^{t+\lfloor\tau_t\rfloor-1}\ell_j(i)\right]\le\frac{1}{\eta}\mathbb{E}\left[\log\left(\frac{W_t}{w_t(i_t^*)}\right)-\log\left(\frac{W_{t+1}}{w_{t+1}(i_{t+1}^*)}\right)\right]$$

$$+\frac{\eta}{2}\mathbb{E}\left[\frac{1}{p_t(r)}\sum_{i\in V}q_t(i)\tau_t\sum_{j=t}^{t+\tau_t-1}\mathbb{E}[\ell_j(i)^2|a_{1:t-1}]\right]+\mathbb{E}[L_{t+1,*}-L_{t,*}]$$

$$\implies$$

$$\mathbb{E}\left[\sum_{i\in V}q_t(i)\sum_{j=t}^{t+\lfloor\tau_t\rfloor-1}\ell_j(i)\right]\le\frac{1}{\eta}\mathbb{E}\left[\log\left(\frac{W_t}{w_t(i_t^*)}\right)-\log\left(\frac{W_{t+1}}{w_{t+1}(i_{t+1}^*)}\right)\right]$$

$$+\frac{\eta}{2}\mathbb{E}\left[\rho\frac{p_t(r)^2\tau^2}{p_t(r)}\sum_{i\in V}q_t(i)\right]+\mathbb{E}[L_{t+1,*}-L_{t,*}].$$

Notice that $\mathbb{E}[L_{T,*}]=\mathbb{E}[\sum_{t=1}^{T'}\frac{\mathbb{I}(a_t=r)}{p_t(r)}\sum_{j=t}^{t+\lfloor\tau_t\rfloor-1}\ell_j(i^*)]=\mathbb{E}[\sum_{t=1}^{T'}\sum_{j=t}^{t+\lfloor\tau_t\rfloor-1}\ell_j(i^*)]$. Summing over $t=1$ through $T$ and using the fact $\log\left(\frac{W_1}{w_1(i^*)}\right)=\log\left(|V|\right)$ we have

$$\mathbb{E}\left[\sum_{t=1}^{T'}\sum_{i\in V}q_t(i)\sum_{j=t}^{t+\lfloor\tau_t\rfloor-1}(\ell_j(i)-\ell_j(i^*))\right]\le\frac{\log\left(|V|\right)}{\eta}+\frac{\eta}{2}\tau\mathbb{E}\left[\rho\sum_{t=1}^{T'}p_t(r)\tau\right]$$

$$\le\frac{\log\left(|V|\right)}{\eta}+T\eta\tau\rho,$$

where $T'$ is the random variable equaling the number of mini-batches. The last inequality in the above follows since $\tau_T\in o(T)$ and from our while loop we know that $\sum_{t=1}^{T'-1}\tau_t\le T$, thus we can bound $\mathbb{E}[\sum_{t=1}^{T'}\tau_t]\le 2T$. Notice that the LHS in the above inequality is almost equal to the expected

regret of our algorithm. We have $q_t(i) \leq p_t(i) - \beta$ and thus the expected regret is bounded by

$$\mathbb{E}\left[\sum_{t=1}^{T} \ell_t(a_t) - \ell_t(a)\right] \leq \frac{\log\left(|V|\right)}{\eta} + T\eta\tau\rho + T\beta.$$

$\square$

**Lemma D.2.** *Algorithm 7 switches between a revealing and a non-revealing action at most $\frac{T}{\tau}$ times in expectation.*

*Proof.* The number of switches can be upper bounded by twice the number of times $a_t$ is equal to $r$. Thus the expected number of switches is bounded by $\mathbb{E}[\sum_{t=1}^{T'} \mathbb{I}(a_t = r)] = \frac{1}{\tau}\mathbb{E}[\sum_{t=1}^{T'} p_t(r)\tau] = \frac{1}{\tau}\mathbb{E}[\sum_{t=1}^{T'} \tau_t] \leq \frac{2T}{\tau}$. $\square$

To finish the proof of Theorem 3.1 we need to verify that the expected regret of Algorithm 7 is the same as the expected regret of Algorithm 1.

**Lemma D.3.** *Algorithm 7 and Algorithm 1 have the same expected regret bound.*

*Proof.* Let $(p_t)_{t=1}^{T}$ be the sequence of random vectors generated by Algorithm 7 and let $(p'_t)_{t=1}^{T}$ be the sequence of random vectors generated by Algorithm 1. First we show by induction that the distribution of $p_t$ is the same as that of $p'_t$. The base case is trivial as $p_1 = p'_1$. To see that the induction step holds we just notice that if we condition on $p_t$ either both algorithms update $p_{t+1}$ and $p'_{t+1}$ because action $r$ was sampled, in which case the updates are exactly the same, or both algorithms do not update $p_{t+1}$, respectively $p'_{t+1}$. Let $a_t$ and $a'_t$ denote the $t$-th action of Algorithm 7 and Algorithm 1 respectively. We now show that $\mathbb{E}[\ell_t(a_t)] = \mathbb{E}[\ell_t(a'_t)]$. Let $X_t$ denote the random variable indicating the last time before $t$ in which action $r$ was played by Algorithm 7 and let $X'_t$ be the random variable indicating the last time before $t$ in which action $r$ was played by Algorithm 1. Since $X_t$ is function of $p_1, \ldots, p_{t-1}$ and $X'_t$ is a function of $p'_1, \ldots, p'_{t-1}$, then $X_t$ and $X'_t$ have the same distribution. Now we can write

$$\mathbb{E}[\ell_t(a_t)] = \sum_{j=1}^{t-1} \mathbb{P}(X_t = j)\mathbb{E}[\ell_t(a_t)|X_t = j] = \sum_{j=1}^{t-1} \mathbb{P}(X_t = j)\mathbb{E}[\sum_{i \in V} p_t(i)\ell_t(i)|X_t = j]$$

$$= \sum_{j=1}^{t-1} \mathbb{P}(X_t = j)\mathbb{E}[\sum_{i \in V} p_{j+1}(i)\ell_t(i)|X_t = j]$$

$$= \sum_{j=1}^{t-1} \mathbb{P}(X_t = j)\mathbb{E}[\sum_{i \in V} p'_{j+1}(i)\ell_t(i)|X'_t = j]$$

$$= \sum_{j=1}^{t-1} \mathbb{P}(X'_t = j)\mathbb{E}[\ell_t(a'_t)|X'_t = j] = \mathbb{E}[\ell_t(a'_t)].$$

$\square$

*Proof of Theorem 3.1.* Lemma D.3 together with Theorem D.1 imply the bound

$$\mathbb{E}\left[\sum_{t=1}^{T} \ell_t(a_t) - \ell_t(a)\right] \leq \tilde{O}\left(\sqrt{\rho}T^{2/3}\right).$$

Lemma D.2 together with the fact that Algorithm 1 can only switch between the revealing action and non-revealing actions imply the bound on number of switches. $\square$

### D.2 Corralling the Star-graph Algorithms

We use a mini-batch version of Algorithm 1 in Agarwal et al. [2016] where each of the base algorithms is Algorithm 1. We note that the greedy algorithm for computing an approximate minimum dominating set gives a natural way to partition the feedback graph $G$ into star graphs. In

particular, whenever the greedy algorithm adds a vertex $v$ to the dominating set, we create a new instance of the star graph algorithm with revealing vertex $v$ and leaf nodes all neighbors of $v$ which have not already been assigned to a star graph algorithm.

**Lemma D.4.** *For any $i \in [|R|]$, Algorithm 3 ensures that:*

$$\mathbb{E}\left[\sum_{t=1}^{T} \ell_t(a_t) - \ell_t(a_t^i)\right] \leq O\left(\frac{\tau |R| \log(T')}{\eta} + T\eta\right) - \mathbb{E}\left[\frac{\tau \rho_{T',i}}{40\eta \log(T')}\right]$$

*Proof.* From the proof of Lemma 13 in Agarwal et al. [2016] it follows that for any $i \in [|R|]$

$$\sum_{t=1}^{T'} \langle p_t - e_i, \widehat{\ell}_t \rangle \leq O\left(\frac{|R| \log(T')}{\eta} + T'\eta\right) + \sum_{t=1}^{T'} \frac{2\widehat{\ell}_t(a_t)}{T'|R|} - \frac{\rho_{T',i}}{40\eta \log(T')}.$$

Notice that by construction we have $\mathbb{E}[\widehat{\ell}_t(a_t)] = \sum_{i \in [|R|]} \frac{1}{\tau} \sum_{j=t}^{t+\tau-1} \ell_j(a_j^i) \leq |R|$. Also notice that $\mathbb{E}[\langle p_t, \widehat{\ell}_t \rangle] = \mathbb{E}[\frac{1}{\tau} \sum_{j=t}^{t+\tau-1} \ell_j(a_j)]$ and $\mathbb{E}[\widehat{\ell}_t(i)] = \frac{1}{\tau} \sum_{j=t}^{t+\tau-1} \ell_t(a_j^i)$. These imply

$$\mathbb{E}\left[\sum_{t=1}^{T'} \frac{1}{\tau} \sum_{j=t}^{t+\tau-1} \ell_j(a_j) - \frac{1}{\tau} \sum_{j=t}^{t+\tau-1} \ell_t(a_j^i)\right] \leq O\left(\frac{|R| \log(T')}{\eta} + T'\eta\right) + \sum_{t=1}^{T'} \frac{2\widehat{\ell}_t(a_t)}{T'|R|} - \frac{\rho_{T',i}}{40\eta \log(T')}.$$

Multiplying by $\tau$ and using the fact that $T'\tau = T$ finishes the proof. $\qquad\square$

The following theorem from Agarwal et al. [2016] shows that restarting the $i$-th algorithm in line 16 of Algorithm 3 does not hinder the regret bound by too much.

**Theorem D.5** (Theorem 15 [Agarwal et al., 2016])**.** *Suppose a base algorithm $B_i$ is such that if the loss sequence $(\ell_t)_{t=1}^T$ is replaced by $\ell'_t = \rho_t \ell_t$ such that $\mathbb{E}[\ell'_t] = \ell_t$, its regret bound changes from $R(T)$ to $\mathbb{E}[\rho^\alpha]R(T)$, where $\rho = \max_{t \leq T} \rho_t$. Let $(a_t^i)_{t \leq T}$ be the action sequence generated by $B_i$ ran under Algorithm 3. Then for any action $a$ in the action set of $B_i$, it holds that*

$$\mathbb{E}\left[\sum_{t=1}^{T} \ell'_t(a_t^i) - \ell'_t(a)\right] \leq \frac{2^\alpha}{2^\alpha - 1} \mathbb{E}[\rho^\alpha]R(T).$$

**Theorem D.6.** *Let $\tau = \frac{T^{1/3}}{|R|^{1/4}}, \eta = \frac{|R|^{1/4}}{40 \log(T')T^{1/3}c \log(|V|)}$, where $c$ is a constant independent of $T$, $\tau$, $|V|$ or $|R|$. For any $a \in V$, Algorithm 3 ensures that:*

$$\mathbb{E}\left[\sum_{t=1}^{T} \ell_t(a_t) - \ell_t(a)\right] \leq \tilde{O}\left(\sqrt{|R|}T^{2/3}\right).$$

*Further the expected number of switches of the algorithm is bounded by $T^{2/3}|R|^{1/3}$.*

*Proof of Theorem 3.3.* For any action $a \in V$, let $i_a$ be the star-graph algorithm which has $a$ in its actions and let its regret be $R_{i_a}(T)$. Notice that the loss estimators $\ell'_t(i) = \frac{\ell_{t+j}(a_{t+j})}{p_t(i_t)}\mathbb{I}\{i = i_t\}$ we feed the algorithm are such that $\mathbb{E}[\ell'_t(i)^2] \leq \rho_T$. Now Theorem 3.1 implies that the condition of Theorem D.5 is satisfied with $\alpha = 1/2$. Thus, Theorem D.5 implies that

$$\mathbb{E}\left[\sum_{t=1}^{T} \ell'_t(a_t) - \ell'_t(a)\right] \leq \sqrt{2}(\sqrt{2} + 1)\mathbb{E}[\rho_{T',i_a}^{1/2}]3T^{2/3} \log(|V|).$$

Combining the above with Lemma D.4 we have

$$\mathbb{E}\left[\sum_{t=1}^{T} \ell_t(a_t) - \ell_t(a)\right] \leq O\left(\frac{\tau |R| \log(T')}{\eta} + T\eta\right) - \mathbb{E}\left[\frac{\tau \rho_{T',i_a}}{40\eta \log(T')}\right] + 3\sqrt{2}(\sqrt{2} + 1)\mathbb{E}[\rho_{T',i_a}^{1/2}]T^{2/3} \log(|V|)$$

Let $c = 3\sqrt{2}(\sqrt{2} + 1)$. We now consider the terms containing $\rho_{T',i_a}$ in the above inequality.

$$c\mathbb{E}[\rho_{T',i_a}^{1/2}]T^{2/3} \log(|V|) - \mathbb{E}\left[\frac{\tau \rho_{T',i_a}}{40\eta \log(T')}\right] = \mathbb{E}\left[\rho_{T',i_a}^{1/2}\left(cT^{2/3} \log(|V|) - \frac{\tau \rho_{T',i_a}^{1/2}}{40\eta \log(T')}\right)\right].$$

Set $\tau = \frac{T^{1/3}}{|R|^{1/4}}, \eta = \frac{|R|^{1/4}}{40 \log(T') T^{1/3} c \log(|V|)}$ to get

$$\mathbb{E}\left[\rho_{T',i_a}^{1/2}\left(cT^{2/3}\log\left(|V|\right) - \frac{\tau \rho_{T',i_a}^{1/2}}{40\eta \log\left(T'\right)}\right)\right] = cT^{2/3}\log\left(|V|\right)\mathbb{E}\left[\rho_{T',i_a}^{1/2}\left(1 - \frac{\rho_{T',i_a}^{1/2}}{|R|^{1/2}}\right)\right]$$

$$\leq c\sqrt{|R|}\log\left(|V|\right)T^{2/3}.$$

Plugging in the the values of $\eta$ and $\tau$ in the rest of the bound finishes the regret bound.

The number of switches is bounded from the fact that Algorithm 3 can switch between star-graph algorithms at most $T^{2/3}|R|^{1/3}$ times and Lemma D.2. $\qquad \square$

## D.3   Improving the Domination Number Dependence for General Feedback Graphs

For convenience of the reader we restate the pseudo code for Algorithm 2 below.

---

**Algorithm 8** Algorithm for general feedback graphs

---

**Input:**  Graph $G(V, E)$, learning rate sequence $(\eta_t)$, exploration rate $\beta \in [0, 1]$, maximum mini-batch $\tau$.
**Output:**  Action sequence $(a_t)_t$.
1: Compute an approximate dominating set $R$
2: $q_1 \equiv Unif(V), u \equiv Unif(R)$
3: **while** $\sum_t \tau_t \leq T$ **do**
4:     $p_t = (1 - \beta)q_t + \beta u$.
5:     Draw $i \sim p_t$, set $\tau_t = p_t(r_i)\tau$, where $r_i$ is the dominating vertex for $i$ and set $a_t = i$.
6:     **if** $a_{t-1} \notin R$ and $a_t \notin R$ **then**
7:         Set $a_t = a_{t-1}$
8:     **end if**
9:     Play $a_t$ for the next $\lfloor \tau_t \rfloor$ iterations.
10:    Set
$$\widehat{\ell}_t(i) = \sum_{j=t}^{t+\lfloor \tau_t \rfloor - 1} \mathbb{I}(a_t = r_i)\frac{\ell_j(i)}{p_t(r_i)}.$$

11:    For all $i \in V$, $q_{t+1}(i) = \frac{q_t(i)\exp\left(-\eta_t\widehat{\ell}_t(i)\right)}{\sum_{j\in V} q_t(j)\exp\left(-\eta_t\widehat{\ell}_t(j)\right)}$.
12:    $t = t + 1$.
13: **end while**

---

**Theorem D.7.** *For any $\beta \geq \frac{|R|}{\tau}$ The expected regret of Algorithm 2 is*

$$\frac{\log\left(|V|\right)}{\eta} + 2\eta\tau T + \beta T.$$

*Further, if the algorithm is augmented similar to Algorithm 7, then it will switch between actions at most $\frac{2T|R|}{\tau}$ times.*

*Proof of Theorem 3.2.*  First note that because of the condition $\beta \geq \frac{|R|}{\tau}$ each of the mini-batches $\lfloor \tau_t \rfloor$ is at least 1, since for any $r \in R$ we have $p_t(r) \geq \frac{\beta}{|R|} \geq \frac{1}{\tau}$, and thus the algorithm will terminate in at most $2T$ iterations. Next, similarly to Lemma D.3, we can analyze the regret of Algorithm 2 by removing lines 6 and 7 when bounding the cumulative loss of the algorithm and then use lines 6 and 7 to guarantee that the algorithm does not switch too often. Let $w_{t+1}(i) = w_t(i)\exp\left(-\eta_t\sum_{j=t}^{t+\lfloor \tau_t\rfloor-1}\mathbb{I}(a_t = r_i)\frac{\ell_j(i)}{p_t(r_i)}\right)$ and $W_t = \sum_{i\in V} w_t(i)$, so that $q_t(i) = \frac{w_t(i)}{W_t}$. Let $V_r$ be the subset of actions dominated by the vertex $r$. Let $i_t^*$ be the best action at time $t$ and let $L_{t,*} = \sum_{s=1}^t \widehat{\ell}_s(i_t^*)$. We consider the difference $\log\left(\frac{W_{t+1}}{w_{t+1}(i_{t+1}^*)}\right) - \log\left(\frac{W_t}{w_t(i_t^*)}\right)$.

$$\log\left(\frac{W_{t+1}}{w_{t+1}(i^*_{t+1})}\right) - \log\left(\frac{W_t}{w_t(i^*_t)}\right) = \eta_t(L_{t+1,*} - L_{t,*})$$

$$+ \log\left(\sum_{r\in R}\sum_{i\in V_r} q_t(i)\exp\left(-\eta_t \sum_{j=t}^{t+\lfloor\tau_t\rfloor-1}\mathbb{I}(a_t = r_i)\frac{\ell_j(i)}{p_t(r_i)}\right)\right)$$

$$\leq \eta_t(L_{t+1,*} - L_{t,*}) - 1$$

$$+ \sum_{r\in R}\sum_{i\in V_r} q_t(i)\exp\left(-\eta_t \sum_{j=t}^{t+\lfloor\tau_t\rfloor-1}\mathbb{I}(a_t = r_i)\frac{\ell_j(i)}{p_t(r_i)}\right)$$

$$\leq \eta_t(L_{t+1,*} - L_{t,*}) - \eta_t\sum_{r\in R}\sum_{i\in V_r} q_t(i)\sum_{j=t}^{t+\lfloor\tau_t\rfloor-1}\mathbb{I}(a_t = r_i)\frac{\ell_j(i)}{p_t(r_i)}$$

$$+ \frac{\eta_t^2}{2}\sum_{r\in R}\sum_{i\in V_r} q_t(i)\left(\sum_{j=t}^{t+\tau_t-1}\mathbb{I}(a_t = r)\frac{\ell_j(i)}{p_t(r)}\right)^2,$$

where the first inequality follows from the fact that $\log\left(() \, x\right) \leq x - 1$ for all $x \geq 0$ and the second inequality follows from the fact that $e^{-x} \leq 1 - x + x^2/2$, for all $x \geq 0$. Set $\eta_t = \eta$ and divide both sides by $\eta$. Shuffling terms around, taking expectation and noting that if one drops the floor function from the quadratic term it will only get larger we arrive at the following

$$\mathbb{E}\left[\sum_{r\in R}\sum_{i\in V_r} q_t(i)\sum_{j=t}^{t+\lfloor\tau_t\rfloor-1}\mathbb{I}(a_t = r)\frac{\ell_j(i)}{p_t(r)} + L_{t+1,*} - L_{t,*}\right]$$

$$\leq \frac{1}{\eta}\mathbb{E}\left[\log\left(\frac{W_t}{w_t(i^*_{r*})}\right) - \log\left(\frac{W_{t+1}}{w_{t+1}(i^*_{r*})}\right)\right] \tag{1}$$

$$+\frac{\eta}{2}\mathbb{E}\left[\sum_{r\in R}\sum_{i\in V_r} q_t(i)\left(\sum_{j=t}^{t+\tau_t-1}\mathbb{I}(a_t = r)\frac{\ell_j(i)}{p_t(r)}\right)^2\right].$$

Consider the term on the LHS.

$$\mathbb{E}\left[\sum_{r\in R}\sum_{i\in V_r} q_t(i)\sum_{j=t}^{t+\lfloor\tau_t\rfloor-1}\mathbb{I}(a_t = r)\frac{\ell_j(i)}{p_t(r)} + L_{t+1,*} - L_{t,*}\right]$$

$$= \mathbb{E}\left[\sum_{r\in R}\sum_{i\in V_r} q_t(i)\sum_{j=t}^{t+\lfloor\tau_t\rfloor-1}\ell_j(i) + L_{t+1,*} - L_{t,*}\right],$$

where in the last inequality we used that $\ell_j(i) \leq 1$ for all $i \in V$. Now we consider the second term on the RHS of the inequality.

$$\mathbb{E}\left[\sum_{r\in R}\sum_{i\in V_r} q_t(i)\left(\sum_{j=t}^{t+\tau_t-1}\mathbb{I}(a_t = r)\frac{\ell_j(i)}{p_t(r)}\right)^2\right]$$

$$=\mathbb{E}\left[\sum_{r\in R}\sum_{i\in V_r} q_t(i)\mathbb{E}\left[\frac{\mathbb{I}(a_t = r)}{p_t(r)^2}\left(\sum_{j=t}^{t+\tau_t-1}\ell_j(i)\right)^2\Big|a_{1:t-1}\right]\right]$$

$$\leq\mathbb{E}\left[\sum_{r\in R}\sum_{i\in V_r} q_t(i)\mathbb{E}\left[\frac{\mathbb{I}(a_t = r)}{p_t(r)^2}\tau_t^2|a_{1:t-1}\right]\right]$$

Consider the term $\mathbb{E}\left[\frac{\mathbb{I}(a_t=r)}{p_t(r)^2}\tau_t^2|a_{1:t-1}\right]$. We have $a_t = r$ with probability $p_t(r)$ and so $\tau_t = p_t(r)\tau$. Otherwise we have $\frac{\mathbb{I}(a_t=r)}{p_t(r)^2}\tau_t^2 = 0$. Thus the RHS is bounded by

$$\mathbb{E}\left[\sum_{r\in R}\sum_{i\in V_r}q_t(i)\left(\sum_{j=t}^{t+\tau_t-1}\mathbb{I}(a_t=r)\frac{\ell_j(i)}{p_t(r)}\right)^2\right]$$

$$\leq\mathbb{E}\left[\sum_{r\in R}\sum_{i\in V_r}q_t(i)\mathbb{E}\left[\frac{\mathbb{I}(a_t=r)}{p_t(r)^2}\tau_t^2|a_{1:t-1}\right]\right] = \mathbb{E}\left[\sum_{r\in R}\sum_{i\in V_r}q_t(i)p_t(r)\tau^2\right]$$

$$=\tau\mathbb{E}\left[\sum_{r\in R}p_t(r)\tau\mathbb{P}\left[\tau_t = p_t(r)\tau\right]\right] = \tau\mathbb{E}[\tau_t].$$

Summing the LHS and RHS of Equation 1 and using our respective bounds, we get:

$$\mathbb{E}\left[\sum_{t=1}^{T'}\sum_{r\in R}\sum_{i\in V_r}q_t(i)\sum_{j=t}^{t+\lfloor\tau_t\rfloor-1}\ell_j(i) - \sum_{j=t}^{t+\lfloor\tau_t\rfloor-1}\ell_j(i_{r^*}^*)\right]$$

$$\leq\frac{\log\left(|V|\right)}{\eta} + \frac{\eta}{2}\tau\mathbb{E}\left[\sum_{t=1}^{T'}\tau_t\right] \leq \frac{\log\left(|V|\right)}{\eta} + \eta\tau T.$$

Next we notice that the LHS is almost the expected regret of the algorithm, except we need to replace $q_t(i)$ by $p_t(i)$. This is done at the cost of an additional $\beta T$ term, since $q_t(r) \leq p_t(r) - \frac{\beta}{|R|}$ for $r \in R$. Finally we upper bound the number of times the algorithm switches by the number of times it samples a revealing arm which is equal to $\mathbb{E}\left[\sum_{t=1}^{T'}\sum_{r\in R}\mathbb{I}(a_t=r)\right]$. To bound this term we do the following

$$2T \geq \mathbb{E}\left[\sum_{t=1}^{T'}\tau_t\right] = \mathbb{E}\left[\sum_{t=1}^{T'}\mathbb{E}\left[\tau_t|p_t\right]\right] = \mathbb{E}\left[\sum_{t=1}^{T'}\sum_{r\in R}p_t(r)\tau\sum_{i\in V_r}p_t(i)\right]$$

$$\geq\mathbb{E}\left[\sum_{t=1}^{T'}\sum_{r\in R}\tau p_t(r)^2\right] = \tau\mathbb{E}\left[\sum_{t=1}^{T'}\sum_{r\in R}p_t(r)^2\right] \geq \frac{\tau}{|R|}\mathbb{E}\left[\sum_{t=1}^{T'}\left(\sum_{r\in R}p_t(r)\right)^2\right]$$

$$\geq\frac{\tau}{|R|}\mathbb{E}\left[\sum_{t=1}^{T'}\left(\mathbb{E}\left[\sum_{r\in R}p_t(r)|a_{1:(t-1)}\right]\right)^2\right] = \frac{\tau}{|R|}\mathbb{E}\left[\sum_{t=1}^{T'}\left(\sum_{r\in R}\mathbb{I}(a_t=r)\right)^2\right]$$

$$=\frac{\tau}{|R|}\mathbb{E}\left[\sum_{t=1}^{T'}\sum_{r\in R}\mathbb{I}(a_t=r)\right],$$

where the second inequality follows from the fact that $\sum_{i\in V_r}p_t(i) \geq p_t(r)$, the third inequality follows from the fact that $(\sum_{r\in R}p_t(r))^2 \leq |R|\sum_{r\in R}p_t(r)^2$ and the fourth inequality follows from Jensen's inequality for conditional expectations. □

## E  Policy Regret Bounds

In this section we assume that we are provided with a feedback graph for losses with memory $m$. We restrict the feedback graph to only have vertices for repeated $m$-tuples of actions in $V$. In particular we can only observe additional feedback for losses of the type $\ell_t(a, a, \ldots, a)$, where $a \in V$. The algorithm for this setting is based on Algorithm 2. The feedback graph we provide to our policy regret algorithm is the same as for the $m$-memory bounded losses, however, each $m$-tuple vertex is replaced by a copy of a single action e.g. the vertex $(a, \ldots, a)$ is replaced by $a$. Next we split the stream of $T$ losses into mini-batches of size $m$ such that $\widehat{\ell}_t(\cdot) = \frac{1}{m}\sum_{j=1}^{m}\ell_{(t-1)m+j}(\cdot)$. Now we would

simply feed the sequence $(\widehat{\ell}_t)_{t=1}^{T/m}$ to Algorithm 2 if it were not for the constraint on the additional feedback. Suppose that between the $t$-th mini-batch and the $t+1$-st mini-batch Algorithm 2 decides to switch actions so that $a_t \neq a_{t+1}$. In this case no additional feedback is available for $\widehat{\ell}_{t+1}(a_{t+1})$ and the algorithm can not proceed as normal. To fix this minor problem, the provided feedback to Algorithm 2 is that the loss of action $a_{t+1}$ was 0 and all actions adjacent to $a_{t+1}$ also incurred 0 loss. This modification can not occur more times than the number of switches Algorithm 2 does. Since the expected number of switches is bounded by $O(\gamma(G)^{1/3}T^{2/3})$, intuitively the modification becomes benign to the total expected regret. Pseudocode for the above algorithm can be found in Algorithm 5.

---

**Algorithm 9** Policy regret with side observations

---

**Input:** Feedback graph $G(V,E)$, learning rate $\eta$, mini-batch size $\tau$, where $\eta$ and $\tau$ are set as in
    Theorem 3.3.
**Output:** Action sequence $(a_t)_t$.
1: Transform feedback graph $G$ from $m$-tuples to actions and initialize Algorithm 2.
2: **for** $t = 1, \ldots, T/m$ **do**
3:      Sample action $a_t$ from $p_t$ generated by Algorithm 2 and play it for the next $m$ rounds.
4:      **if** $a_{t-1} == a_t$ **then**
5:          Observe mini-batched loss $\widehat{\ell}_t(a_t) = \frac{1}{m}\sum_{j=1}^m \ell_{(t-1)m+j}(a_t)$ and additional side observa-
        tions. Feed mini-batched loss and additional side observations to Algorithm 2.
6:      **else**
7:          Set $\widehat{\ell}_t(a_t) = 0$ and set additional feedback losses to 0. Feed losses to Algorithm 2.
8:      **end if**
9: **end for**

---

**Theorem E.1.** *The expected policy regret of Algorithm 5 is bounded by* $\tilde{O}(m^{1/3}\gamma(G)^{1/3}T^{2/3})$.

*Proof of Theorem 4.1.* Theorem 3.2 guarantees that

$$\mathbb{E}\left[\sum_{t=1}^{T/m}\widehat{\ell}_t(a_t) - \sum_{t=1}^{T/m}\widehat{\ell}_t(a)\right] \leq \tilde{O}\left(\gamma(G)^{1/3}(T/m)^{2/3}\right),$$

for any action $a$. On the other hand we have

$$\mathbb{E}\left[\sum_{t=1}^{T/m}\widehat{\ell}_t(a_t) - \sum_{t=1}^{T/m}\widehat{\ell}_t(a)\right] \leq \mathbb{E}\left[\sum_{t=1}^{T/m}\widehat{\ell}_t(a_t) - \sum_{t=1}^{T/m}\frac{1}{m}\sum_{j=1}^m \ell_{(t-1)m+j}(a)\right]$$

$$=\mathbb{E}\left[\sum_{t=1}^{T/m}\frac{1}{m}\sum_{j=1}^m \ell_{(t-1)m+j}(a_t) - \sum_{t=1}^{T/m}\frac{1}{m}\sum_{j=1}^m \ell_{(t-1)m+j}(a) - \sum_{t=1}^{T/m}\mathbb{I}(a_{t-1} \neq a_t)\frac{1}{m}\sum_{j=1}^m \ell_{(t-1)m+j}(a_t)\right].$$

Combined with the regret bound, the above implies

$$\frac{1}{m}\mathbb{E}[R(T)] \leq \tilde{O}\left(\gamma(G)^{1/3}(T/m)^{2/3}\right) + \mathbb{E}\left[\sum_{t=1}^{T/m}\mathbb{I}(a_{t-1} \neq a_t)\right]. \tag{2}$$

The second term in the right hand side bounded by the number of switches bound in Theorem 3.3 as

$$\mathbb{E}\left[\sum_{t=1}^{T/m}\mathbb{I}(a_{t-1} \neq a_t)\right] \leq \tilde{O}(\gamma(G)^{1/3}(T/m)^{2/3}).$$

Multiplying Inequality 2 by $m$ on both sides finishes the proof. $\qquad\square$

# F    Lower Bound for Non-complete Graphs

Before proceeding with the proof of Theorem 5.1, we introduce the stochastic process defined in Dekel et al. [2014].

**Stochastic process definition:** We denote by $\xi_{1:T}$ a sequence of i.i.d. zero-mean Gaussian random variables with variance $\sigma^2$ and $\rho : [T] \to \{0\} \bigcup [T]$ the parent function, which assigns to $t \in [T]$ a parent $\rho(t) \in [T]$ with $\rho(t) < t$. The stochastic process $W_t$ associated with $\rho(t)$ is defined as

$$\begin{aligned} W_0 &= 0 \\ W_t &= W_{\rho(t)} + \xi_t. \end{aligned} \tag{3}$$

The set of ancestors of $t$ is the set $\rho^*(t) = \rho^*(\rho(t)) \bigcup \{\rho(t)\}$ with $\rho^*(0) = \{\}$. The depth of $\rho$ is $d(\rho) = \max_{t \in [T]} |\rho^*(t)|$. The cut of $\rho$ is $cut(t) = \{s \in [T] : \rho(s) < t \le s\}$ i.e. the set of rounds which are separated from their parent by $t$. The width of $\rho$ is defined as $\omega(\rho) = \max_{t \in [T]} |cut(t)|$. The specific random walk which Dekel et al. [2014] consider has both depth and width logarithmic in $T$. In particular the parent function is defined as

$$\rho(t) = t - 2^{\delta(t)}, \text{where}, \delta(t) = \max\{i \ge 0 : t \equiv 0 \bmod 2^i\} \tag{4}$$

Let us consider two examples of a stochastic processes defined by Equation 3. The first one is just setting $\rho(t) = 0$, so that $W_t$ is just a standard Gaussian variable. The width of this process is just $T$ and its depth is 1. While we have good concentration guarantees over the maximum value of $W_t$ uniformly over all $t \in [T]$, which is important for controlling the losses, it is very easy to gain information about actions 1 and 2 without switching. Indeed one can just first play 1 for a sufficient number of iteration and then play 2 for fixed number of iterations to be able, with high probability, to distinguish between the two losses. Now consider a Gaussian random walk where $\rho(t) = t - 1$. In this case the cut is 1 but the depth is $T$. It turns out that to distinguish between two processes with small width, we require that we observe both the processes at the same time (or times differing by a small amount). This is intuitively because of the large drift of the process that occurs between $W_t$ and $W_{t+k}$. We note that the simple Gaussian walk is not a good process for the losses, since its depth is too large for us to be able to control the size of the (unclipped) losses.

The feedback graph we work for the reset of this section is $G(V, E)$, where $V = \{1, 2, 3\}$ and $E = \{(1, 3), (2, 3), (1, 1), (2, 2), (3, 3)\}$ (see Figure 2).

**Constructing the losses:** We consider the following adversarial sequence of losses. First sample an action uniformly at random from $\{1, 2\}$. WLOG we condition on the event that the sampled action is 1. Next set $\ell_t(3) = 1$, $\ell_t(2) = clip(W_t + \frac{1}{2})$, $\ell_t(1) = clip(W_t + \frac{1}{2} - \epsilon)$, where $clip(\alpha) = \min\{\max\{\alpha, 0\}, 1\}$. The intuition behind our lower bound is very simple and holds for a general feedback graph. It is as follows: if we do not have a complete feedback graph then there are at least two actions which do not tell us anything about each other. We leverage this by selecting one of the two actions uniformly at random to be the *best* action. If we play an action which is not 1 or 2 we incur constant regret in that turn but we can gain information about the losses of both 1 and 2. If we play 2, then we do not learn anything about 1 and if we play 1 we do not learn anything about 2. In these two cases the per round regret incurred is $\epsilon$, however, because of the loss construction, we need to switch between these actions to be able to distinguish them and thus we will incur regret from switching. Overall the loss construction together with the result in Dekel et al. [2014] implies that to distinguish between 1 and 2 we need to observe the losses of both actions at the same time or switch between them at least $\tilde{\Omega}(T^{2/3})$ rounds. This is what we formally argue below.

Let $Y_t$ be the observed loss vector associated with the action at time $t$, $a_t$, i.e. if $a_t = 2$ then $Y_t = W_t + \frac{1}{2}$, if $a_t = 1$ then $Y_t = W_t + \frac{1}{2} - \epsilon$ and if $a_t = 3$ then $Y_t = \begin{pmatrix} W_t + \frac{1}{2} \\ W_t + \frac{1}{2} - \epsilon \end{pmatrix}$. We let $Y_0 = 1/2$. We let $\mathcal{Q}_1$ be the probability measure on the $\sigma$-field $\mathcal{F}$ generated by $\{Y_t\}_{t=0}^T$. Let $\mathcal{Q}_0$ be the probability measure on the same $\sigma$-field if $\ell_t(1) = \ell_t(2) = clip(W_t + \frac{1}{2})$ i.e. there is no best action. In this case $Y_t = W_t + \frac{1}{2}$ for $a_t = 1$ or $a_t = 2$ and $Y_t = \begin{pmatrix} W_t + \frac{1}{2} \\ W_t + \frac{1}{2} \end{pmatrix}$ if $a_t = 2$. Denote by $d_{\text{TV}}^{\mathcal{F}}(\mathcal{Q}_0, \mathcal{Q}_1)$ the total variational distance between $\mathcal{Q}_0$ and $\mathcal{Q}_1$ on the $\sigma$-field $\mathcal{F}$. Let $D_{\text{KL}}(\mathcal{Q}_0||\mathcal{Q}_1)$ be the KL-divergence between $\mathcal{Q}_0$ and $\mathcal{Q}_1$. We now show that a sufficiently large number of switches between actions 1 and 2 or choosing action 3 is required to distinguish between $\mathcal{Q}_0$ and $\mathcal{Q}_1$. As it was discussed above, the width of the process plays an important role, which is clarified by the lemma below. It essentially is an upper bound on the number of switches required to distinguish between $\mathcal{Q}_0$ and $\mathcal{Q}_1$.

**Lemma F.1.** *Let $M$ be the number of times the player's strategy switched between actions $1$ and $2$. Let $N$ be the number of times the payer chose to play action $3$. Then $\mathrm{d}_{TV}^{\mathcal{F}}\left(\mathcal{Q}_0, \mathcal{Q}_1\right) \leq \frac{\epsilon}{2\sigma}\sqrt{\omega(\rho)\mathbb{E}_{\mathcal{Q}_0}[M+N]}$.*

*Proof.* Let $Y_{0:t}$ denote $(Y_0, Y_1, \ldots, Y_t)$ and whenever $Y_t$ is a vector, let $Y_t(i)$ be its $i$-th coordinate. We assume that the player is deterministic. By Yao's minimax principle this is without loss of generality. Thus we have that $a_t$ is a deterministic function of $Y_{0:t-1}$. Using the chain rule for relative entropy and by the construction of $W_t$, we have:

$$\mathrm{D}_{\mathrm{KL}}\left(\mathcal{Q}_0(Y_{0:T})||\mathcal{Q}_1(Y_{0:T})\right) = \mathrm{D}_{\mathrm{KL}}\left(\mathcal{Q}_0(Y_0)||\mathcal{Q}_1(Y_1)\right) + \sum_{t=1}^{T} \mathrm{D}_{\mathrm{KL}}\left(\mathcal{Q}_0(Y_t|Y_{\rho*(t)})||\mathcal{Q}_1(Y_t|Y_{\rho*(t)})\right).$$

Let us consider the term $\mathrm{D}_{\mathrm{KL}}\left(\mathcal{Q}_0(Y_t|Y_{\rho*(t)})||\mathcal{Q}_1(Y_t|Y_{\rho*(t)})\right)$. First assume that $a_t = a_{\rho(t)} \neq 3$. Then $Y_t = \mathcal{N}(Y_{\rho(t)}, \sigma^2)$ under both $\mathcal{Q}_0$ and $\mathcal{Q}_1$. Next consider the case when $a_t = a_{\rho(t)} = 3$. In this case $Y_t = \mathcal{N}\left(\begin{pmatrix} Y_{\rho(t)}(2) \\ Y_{\rho(t)}(2) \end{pmatrix}, \sigma^2 I_2\right)$ under $\mathcal{Q}_0$ and $Y_t = \mathcal{N}\left(\begin{pmatrix} Y_{\rho(t)}(2) - \epsilon \\ Y_{\rho(t)}(2) \end{pmatrix}, \sigma^2 I_2\right)$ under $\mathcal{Q}_1$. If $a_t \neq a_{\rho(t)}$ we have 6 options:

1. $a_{\rho(t)} = 3$

   (a) $a_t = 1$, in this case $Y_t = \mathcal{N}(Y_{\rho(t)}(2), \sigma^2)$ under $\mathcal{Q}_0$ and $Y_t = \mathcal{N}(Y_{\rho(t)}(2) - \epsilon, \sigma^2)$ under $\mathcal{Q}_1$;

   (b) $a_t = 2$ in this case $Y_t = \mathcal{N}(Y_{\rho(t)}(2), \sigma^2)$ under $\mathcal{Q}_0$ and $Y_t = \mathcal{N}(Y_{\rho(t)}(2), \sigma^2)$ under $\mathcal{Q}_1$;

2. $a_{\rho(t)} = 1$

   (a) $a_t = 3$, in this case $Y_t = \mathcal{N}\left(\begin{pmatrix} Y_{\rho(t)} \\ Y_{\rho(t)} \end{pmatrix}, \sigma^2 I_2\right)$ under $\mathcal{Q}_0$ and $Y_t = \mathcal{N}\left(\begin{pmatrix} Y_{\rho(t)} \\ Y_{\rho(t)} + \epsilon \end{pmatrix}, \sigma^2 I_2\right)$ under $\mathcal{Q}_1$;

   (b) $a_t = 2$ in this case $Y_t = \mathcal{N}(Y_{\rho(t)}, \sigma^2)$ under $\mathcal{Q}_0$ and $Y_t = \mathcal{N}(Y_{\rho(t)} + \epsilon, \sigma^2)$ under $\mathcal{Q}_1$;

3. $a_{\rho(t)} = 2$

   (a) $a_t = 3$, in this case $Y_t = \mathcal{N}\left(\begin{pmatrix} Y_{\rho(t)} \\ Y_{\rho(t)} \end{pmatrix}, \sigma^2 I_2\right)$ under $\mathcal{Q}_0$ and $Y_t = \mathcal{N}\left(\begin{pmatrix} Y_{\rho(t)} - \epsilon \\ Y_{\rho(t)} \end{pmatrix}, \sigma^2 I_2\right)$ under $\mathcal{Q}_1$;

   (b) $a_t = 1$ in this case $Y_t = \mathcal{N}(Y_{\rho(t)}, \sigma^2)$ under $\mathcal{Q}_0$ and $Y_t = \mathcal{N}(Y_{\rho(t)} - \epsilon, \sigma^2)$ under $\mathcal{Q}_1$.

Thus we have

$$\begin{aligned} \mathrm{D}_{\mathrm{KL}}\left(\mathcal{Q}_0(Y_t|Y_{\rho*(t)})||\mathcal{Q}_1(Y_t|Y_{\rho*(t)})\right) &= \mathcal{Q}_0(a_t = a_{\rho(t)} = 3)\mathrm{D}_{\mathrm{KL}}\left(\mathcal{N}(0, \sigma^2)||\mathcal{N}(-\epsilon, \sigma^2)\right) \\ &\quad + \mathcal{Q}_0(a_{\rho(t)=3}, a_t = 1)\mathrm{D}_{\mathrm{KL}}\left(\mathcal{N}(0, \sigma^2)||\mathcal{N}(-\epsilon, \sigma^2)\right) \\ &\quad + \mathcal{Q}_0(a_{\rho(t)=1}, a_t = 3)\mathrm{D}_{\mathrm{KL}}\left(\mathcal{N}(0, \sigma^2)||\mathcal{N}(\epsilon, \sigma^2)\right) \\ &\quad + \mathcal{Q}_0(a_{\rho(t)=1}, a_t = 2)\mathrm{D}_{\mathrm{KL}}\left(\mathcal{N}(0, \sigma^2)||\mathcal{N}(\epsilon, \sigma^2)\right) \\ &\quad + \mathcal{Q}_0(a_{\rho(t)=2}, a_t = 3)\mathrm{D}_{\mathrm{KL}}\left(\mathcal{N}(0, \sigma^2)||\mathcal{N}(-\epsilon, \sigma^2)\right) \\ &\quad + \mathcal{Q}_0(a_{\rho(t)=2}, a_t = 1)\mathrm{D}_{\mathrm{KL}}\left(\mathcal{N}(0, \sigma^2)||\mathcal{N}(-\epsilon, \sigma^2)\right) \\ &= \frac{\epsilon^2}{2\sigma^2}\mathcal{Q}_0(A_t), \end{aligned}$$

where $A_t$ is the event that either action 3 was played at round $t$ or there were odd number of switches between actions 1 and 2. Let $N$ denote the random number of times action 3 was played and let $M$

denote the random number of switches between action 1 and action 2. Let $S_{1:M}$ denote the random sequence of times during which there was a switch. Then we have

$$\sum_{t=1}^{T} \mathbb{1}_{A_t} \leq \sum_{r=1}^{M} \sum_{t \in \mathrm{cut}(S_r)} \mathbb{1}_{A_t} + N \leq \omega(\rho)(M+N),$$

where $\mathrm{cut}(t)$ and $\omega(\rho)$ are defined in Dekel et al. [2014]. Thus

$$D_{\mathrm{KL}}\left(\mathcal{Q}_0(Y_t|Y_{\rho*(t)})||\mathcal{Q}_1(Y_t|Y_{\rho*(t)})\right) \leq \frac{\epsilon^2 \omega(\rho)}{2\sigma^2} \mathbb{E}_{\mathcal{Q}_0}[M+N].$$

Pinsker's inequality that $\mathrm{d}_{\mathrm{TV}}^{\mathcal{F}}(\mathcal{Q}_0, \mathcal{Q}_1) \leq \frac{\epsilon}{2\sigma}\sqrt{\omega(\rho)\mathbb{E}_{\mathcal{Q}_0}[M+N]}$ $\qquad\square$

Next we show that, because of the depth of the random walk, we are able to say that with high probability most of the non-clipped losses will be equal to the clipped losses. The implications of this result are two-fold. First the regret incurred on the non-clipped versions is close to the regret incurred on the clipped version. Secondly, we are able to say that loss of action 3 is worse by a constant from the losses of actions 1 and 2 often enough, so that we also incur constant regret when playing action 3 as compared to the other two actions. Let $\ell_t'$ denote the non-clipped version of $\ell_t$ and define

$$R' = \sum_{t=1}^{T} \ell_t'(a_t) + M - \min_{a \in \mathcal{A}} \sum_{t=1}^{T} \ell_t'(a)$$

$$R = \sum_{t=1}^{T} \ell_t(a_t) + M - \min_{a \in \mathcal{A}} \sum_{t=1}^{T} \ell_t(a)$$

Lemma 4 in Dekel et al. [2014] compares $R'$ to $R$

**Lemma F.2.** *For $T \geq 6$, $\mathbb{E}[R] \geq \mathbb{E}[R'] - \epsilon T/6$.*

We now lower bound $\mathbb{E}[R']$.

**Lemma F.3.** *Let $\mathcal{Q}_2$ be the conditional distribution induced by sampling the best action to be equal to $2$. Then*

$$\mathbb{E}[R'] \geq \frac{\epsilon T}{2} - \frac{\epsilon T}{2}(\mathrm{d}_{TV}^{\mathcal{F}}(\mathcal{Q}_0, \mathcal{Q}_1) + \mathrm{d}_{TV}^{\mathcal{F}}(\mathcal{Q}_0, \mathcal{Q}_2)) + \mathbb{E}\left[M + \frac{N}{7}\right]$$

*Proof.* First let us consider the amount of regret the player incurs for picking action 3 N times. To do this we consider the number of times $1/2 + W_t > 5/6$. The expected number of times this occurs is

$$\mathbb{E}\sum_{t=1}^{T} \mathbb{I}(1/2 + W_t > 5/6) \leq \sum_{t=1}^{T} \mathbb{P}\left(|W_t| + \frac{1}{2} \geq \frac{5}{6}\right) \leq \sum_{t=1}^{T} e^{-\frac{1}{d(\rho)\sigma^2}} \leq \sum_{t=1}^{T} e^{-\frac{9\log(T)}{2}} \leq 1.$$

Thus in expectation the regret for picking action 2 N times is at least $(1/6 + \epsilon)(N-1)$. Since we choose $\epsilon = \tilde{\Theta}(T^{-1/3})$, for sufficiently large $T$ we have that in expectation the regret for picking action 3 N times is at least $(N-1)/6$. Let $\chi$ denote the uniform random variable over actions $\{1,2\}$, which picks the best action in the beginning of the game. Denote by $B_i$ the number of times action $i$ was played. Then $\mathbb{E}[R'] \geq \mathbb{E}[\epsilon(T - N - B_\chi) + M + (N-1)/6]$ (this is a lower bound since $M$ only tracks the switches between actions 1 and 2, so the switches to and from action 2 are left out). Thus we have

$$\mathbb{E}[R'] = \frac{\mathbb{E}[\epsilon(T - N - B_1) + M + (N-1)/6|\chi = 1] + \mathbb{E}[\epsilon(T - N - B_2) + M + (N-1)/6|\chi = 2]}{2}$$

$$= \epsilon T - \frac{\epsilon}{2}\left(\mathbb{E}_{\mathcal{Q}_1}[B_1] + \mathbb{E}_{\mathcal{Q}_2}[B_0]\right) + \mathbb{E}\left[M + \frac{N-1}{6} - \epsilon N\right].$$

Since $\epsilon = \tilde{\Theta}(T^{-1/3})$ we have $\frac{N-1}{6} - \epsilon N \leq \frac{N}{7}$. Consider $\mathbb{E}_{\mathcal{Q}_1}[B_1]$, we have

$$\mathbb{E}_{\mathcal{Q}_1}[B_1] - \mathbb{E}_{\mathcal{Q}_0}[B_1] = \sum_{t=1}^{T} (\mathcal{Q}_1(a_t = 1) - \mathcal{Q}_0(a_t = 1)) \leq T\mathrm{d}_{\mathrm{TV}}^{\mathcal{F}}(\mathcal{Q}_0, \mathcal{Q}_1).$$

A similar inequality holds for $\mathbb{E}_{\mathcal{Q}_2}[N_0]$ and thus we get

$$\mathbb{E}_{\mathcal{Q}_1}[B_1] + \mathbb{E}_{\mathcal{Q}_2}[B_0] \leq T(\mathrm{d}_{\mathrm{TV}}^{\mathcal{F}}(\mathcal{Q}_0, \mathcal{Q}_1) + \mathrm{d}_{\mathrm{TV}}^{\mathcal{F}}(\mathcal{Q}_0, \mathcal{Q}_2)) + \mathbb{E}_{\mathcal{Q}_0}[B_0 + B_1]$$
$$\leq T(\mathrm{d}_{\mathrm{TV}}^{\mathcal{F}}(\mathcal{Q}_0, \mathcal{Q}_1) + \mathrm{d}_{\mathrm{TV}}^{\mathcal{F}}(\mathcal{Q}_0, \mathcal{Q}_2)) + T - \mathbb{E}_{\mathcal{Q}_0}[N].$$

The above implies

$$\mathbb{E}[R'] \geq \frac{\epsilon T}{2} - \frac{\epsilon T}{2}(\mathrm{d}_{\mathrm{TV}}^{\mathcal{F}}(\mathcal{Q}_0, \mathcal{Q}_1) + \mathrm{d}_{\mathrm{TV}}^{\mathcal{F}}(\mathcal{Q}_0, \mathcal{Q}_2)) + \mathbb{E}\left[M + \frac{N}{7}\right] + \frac{\epsilon}{2}\mathbb{E}_{\mathcal{Q}_0}[N].$$

$\square$

Putting the above two lemmas together, we are able to show the following result.

**Theorem F.4.** *For any non-complete feedback graph $G$, there exists a sequence of losses on which any algorithm $\mathcal{A}$ in the informed setting incurs expected regret at least*

$$R_T(\mathcal{A}) \geq \Omega\left(\frac{T^{2/3}}{\log(T)}\right).$$

*Proof of Theorem 5.1.* First assume that the event $M + N/7 > \epsilon T$ does not occur on losses generated from $\mathcal{Q}_0$ or $\mathcal{Q}_i$. This implies $\mathcal{Q}_0(M + N/7 > \epsilon T) = \mathcal{Q}_i(M + N/7 > \epsilon T) = 0$. Then

$$\mathbb{E}_{\mathcal{Q}_0}[M + N/7] - \mathbb{E}[M + N/7] = \frac{\mathbb{E}_{\mathcal{Q}_0}[M + N/7] - \mathbb{E}_{\mathcal{Q}_1}[M + N/7] + \mathbb{E}_{\mathcal{Q}_0}[M + N/7] - \mathbb{E}_{\mathcal{Q}_2}[M + N/7]}{2}$$
$$\leq \frac{\epsilon T}{2}(\mathrm{d}_{\mathrm{TV}}^{\mathcal{F}}(\mathcal{Q}_0, \mathcal{Q}_1) + \mathrm{d}_{\mathrm{TV}}^{\mathcal{F}}(\mathcal{Q}_0, \mathcal{Q}_2)).$$

The above, together with Lemma F.3 implies

$$\mathbb{E}[R'] \geq \frac{\epsilon T}{2} - \epsilon T(\mathrm{d}_{\mathrm{TV}}^{\mathcal{F}}(\mathcal{Q}_0, \mathcal{Q}_1) + \mathrm{d}_{\mathrm{TV}}^{\mathcal{F}}(\mathcal{Q}_0, \mathcal{Q}_2)) + \mathbb{E}_{\mathcal{Q}_0}\left[M + \frac{N}{7}\right].$$

Applying Lemma F.2 now gives

$$\mathbb{E}[R] \geq \frac{\epsilon T}{3} - \epsilon T(\mathrm{d}_{\mathrm{TV}}^{\mathcal{F}}(\mathcal{Q}_0, \mathcal{Q}_1) + \mathrm{d}_{\mathrm{TV}}^{\mathcal{F}}(\mathcal{Q}_0, \mathcal{Q}_2)) + \mathbb{E}_{\mathcal{Q}_0}\left[M + \frac{N}{7}\right].$$

On the other hand we can bound $(\mathrm{d}_{\mathrm{TV}}^{\mathcal{F}}(\mathcal{Q}_0, \mathcal{Q}_1) + \mathrm{d}_{\mathrm{TV}}^{\mathcal{F}}(\mathcal{Q}_0, \mathcal{Q}_2))/2$ by Lemma F.1 as

$$(\mathrm{d}_{\mathrm{TV}}^{\mathcal{F}}(\mathcal{Q}_0, \mathcal{Q}_1) + \mathrm{d}_{\mathrm{TV}}^{\mathcal{F}}(\mathcal{Q}_0, \mathcal{Q}_2))/2 \leq \frac{\epsilon}{\sigma\sqrt{2}}\sqrt{\mathbb{E}_{\mathcal{Q}_0}[M + N]\log(T)}.$$

This implies

$$\mathbb{E}[R] \geq \frac{\epsilon T}{3} - \frac{\sqrt{2}\epsilon^2 T}{\sigma}\sqrt{\mathbb{E}_{\mathcal{Q}_0}[M + N]\log(T)} + \mathbb{E}_{\mathcal{Q}_0}\left[M + \frac{N}{7}\right].$$

Let $x = \sqrt{\mathbb{E}_{\mathcal{Q}_0}[M + N]}$. Then we have

$$\mathbb{E}[R] \geq \frac{\epsilon T}{3} - \frac{\sqrt{2}\epsilon^2 T\sqrt{\log(T)}}{\sigma}x + \frac{x^2}{7}.$$

The quadratic $\frac{x^2}{7} - \frac{\sqrt{2}\epsilon^2 T\sqrt{\log(T)}}{\sigma}x$ has minimum $-\frac{7\log(T)\epsilon^4 T^2}{2\sigma^2}$. We set $\epsilon = c\frac{1}{T^{1/3}\log(T)}$ for a constant $c$ to be determined later. We then have

$$\mathbb{E}[R] \geq \frac{cT^{2/3}}{3\log(T)} - \frac{7c^4}{2}\frac{T^{2/3}}{\log(T)^3\sigma^2}.$$

Set $\sigma = \frac{1}{\log(T)}$. The above implies

$$\mathbb{E}[R] \geq \frac{T^{2/3}}{\log(T)}\left(\frac{c}{3} - \frac{7c^4}{2}\right).$$

Choosing $c = \frac{1}{42^{1/3}}$ gives $\frac{c}{3} - \frac{7c^4}{2} \geq \frac{1}{16}$.

Suppose there is some strategy for which $M + N/7 \geq c\frac{T^{2/3}}{\log(T)}$ occurs. Let this strategy have regret $R$. We change the strategy in the following way. Keep track of $M + N/7$ and the moment it exceeds $c\frac{T^{2/3}}{\log(T)}$ pick an action which has had loss smaller than $5/6$. If there is no such action, pick any action and play it until the end of the game. With probability at least $1/T$ we know that such an action exists and that it was set according to the stochastic process construction. Thus the regret of the new strategy $R^*$ is bounded by $\mathbb{E}[R^*] \leq \mathbb{E}[R] + (1 - 1/T)\epsilon T + 1/T \times T \leq 2\mathbb{E}[R] + 1$. Since the lower bound holds for $\mathbb{E}[R^*]$ the proof is complete. $\qquad\square$

## G   Lower Bound for Disjoint Union of Star Graphs

Let $G$ be the graph which is a union of star graphs. Let $R$ be the set of revealing vertices for the star graphs. We denote by $V_i$ the set of vertices associated with the star graph with revealing vertex $v_i$. First for each star graph we sample an *active* vertex uniformly at random from its leaves. Next we sample the best vertex uniformly at random from the set of active vertices. We set the loss of the best vertex to be $clip(W_t + 1/2 - \epsilon)$ and the loss of all other active vertices to $clip(W_t + 1/2)$. For any star graph consisting of a single vertex, we treat the vertex as a leaf. The following theorem follows as an easy reduction from the proof of Dekel et al. [2014].

**Theorem G.1.** *The expected regret of any algorithm $\mathcal{A}$ on a disjoint union of star graphs is lower bounded as follows:*

$$R_T(\mathcal{A}) \geq \Omega\left(\frac{\gamma(G)^{1/3}T^{2/3}}{\log(T)}\right).$$

*Proof of Theorem 5.2.* Let $\mathcal{I}$ be the set of all possible ways to sample a set of active vertices. Let $\mathbb{E}_i$ be the expectation conditioned on the event that the set of active vertices indexed by $i \in \mathcal{I}$ is sampled in the beginning of the game. Consider the subgraph induced by the active vertices $I$ and all of their neighbors $R$. Suppose that there exists a player's strategy such that $\mathbb{E}_i[R] \leq o\left(\frac{\gamma(G)^{1/3}T^{2/3}}{\log(T)}\right)$. We claim this strategy implies a regret upper bound for bandits with switching costs of the order $o\left(\frac{\gamma(G)^{1/3}T^{2/3}}{\log(T)}\right)$. We convert the player's strategy over $I \bigcup R$ to a strategy over $I$. For every time that $a_t \in R$ is played, we replace $a_t$ by the *unique* neighbor of $a_t$ in $I$. This updated strategy's regret is at most the regret of the original strategy and thus by our assumption it has regret at most $o\left(\frac{\gamma(G)^{1/3}T^{2/3}}{\log(T)}\right) = \left(\frac{|I|^{1/3}T^{2/3}}{\log(T)}\right)$. This is in contradiction with the result of Dekel et al. [2014] since the subgraph induced by $I$ is precisely modeling bandit feedback and the losses of actions in $I$ are exactly constructed as in Dekel et al. [2014]. Thus we have $\mathbb{E}[R] \geq \frac{1}{|\mathcal{I}|}\sum_{i \in \mathcal{I}} \mathbb{E}_i[R] = \tilde{\Omega}\left(\frac{\gamma(G)^{1/3}T^{2/3}}{\log(T)}\right).$ $\qquad\square$

Even though the above theorem is a trivial consequence of the result in Dekel et al. [2014] it can also be proved in another way. Let $I$ denote the set of conditional distributions induced by the observed losses, where the conditioning is with respect to the random sampling of vertices as described in the beginning of the section. The general idea of the complicated proof is to count the number of distributions which each strategy of the player gains information about. For example a strategy which switches between two revealing vertices $v_i$ and $v_j$ will gain information about $deg(v_i)deg(v_j)$ distributions. Now the lower bound follows from a careful counting of the number of distributions for which we gain information by switching between revealing vertices. This counting argument can be generalized beyond union of star graphs, by considering an appropriate pair of minimal dominating/maximal independent sets. We leave a detailed argument for future work.

### G.1   Counting Argument for Theorem 5.2

Let $\mathcal{I}$ denote the set of all possible ways to sample active vertices. The cardinality of this set is $|\mathcal{I}| = \prod_{v_i \in R} deg(v_i)$. Denote by $\mathcal{Q}_0^i$ the conditional distribution generated by the observed losses if all losses for active vertices indexed by $i \in \mathcal{I}$ were set to $clip(W_t + 1/2)$. Denote by $\mathcal{Q}_j^i$ the

conditional distribution generated by the observed losses when active vertex $j$ is chosen to be the best given the active vertices are indexed by $i \in \mathcal{I}$. Let $M_j^i$ denote the random variable counting the number of times the player switched from and to an action adjacent to $j$. Let $N_j^i$ denote the random variable counting the number of times the player played an action adjacent to $j$.

**Lemma G.2.** *For all $i \in \mathcal{I}$ and $j \in [|R|]$ it holds that $\mathrm{d}_{TV}^{\mathcal{F}}\left(\mathcal{Q}_0^i, \mathcal{Q}_j^i\right) \leq \frac{\epsilon}{2\sigma}\sqrt{\omega(\rho)\mathbb{E}_{\mathcal{Q}_0^i}[M_j^i + N_j^i]}$.*

*Proof.* Fix $i \in \mathcal{I}$. Repeat the proof of Lemma H.1. Due to the construction of the losses we have $|I_i^*|\phi(G_i) = 1$, where $G_i$ is the induced subgraph of $G$ by the active vertices and the revealing set $R$ and $I_i^*$ is the set of active vertices. The result follows. □

Let $M_i$ denote the random variable measurable with respect to the draw of $i \in \mathcal{I}$ which counts the total number of switches. Similarly let $N_i$ count the total number of times a revealing vertex of degree at least 2 was played.

**Lemma G.3.** *The following holds*

$$\frac{1}{|R||\mathcal{I}|}\sum_{i \in \mathcal{I}}\sum_{j \in [|R|]}\mathrm{d}_{TV}^{\mathcal{F}}\left(\mathcal{Q}_0^i, \mathcal{Q}_j^i\right) \leq \frac{\epsilon}{\sigma\sqrt{2|R|}}\sqrt{\frac{\omega(\rho)}{|\mathcal{I}|}\sum_{i \in \mathcal{I}}\mathbb{E}_{\mathcal{Q}_0^i}[M_i + N_i]}.$$

*Proof.* Notice that conditioned on the draw of $i \in \mathcal{I}$ we have $\sum_{j \in [|R|]} N_i^j \leq N_i$. This happens because there is only one revealing vertex adjacent to the best vertex for every $\mathcal{Q}_i^j$, i.e., the revealing vertex indexed by $j \in [|R|]$. Similarly we have $\sum_{j \in [|R|]} M_i^j \leq 2M_i$, where the constant two appears because we have counted each switch twice – once from action $j$ and once to action $j$. Using Lemma G.2 with concavity of the square root finishes the proof. □

The above lemma was easy to prove because we did not have two vertices which are dominated simultaneously by two different neighbors in $R$. This allowed us to count very easily the number of times we might have over-count $N_i$ for two different choices of the best action. We were also lucky that it was impossible to gain information about the best action proportional to the degree of a revealing vertex. For a general graph both of these events can happen and the counting argument would have to be more careful. Indeed we expect to see a factor similar to $\phi(G)$, which appeared in Lemma H.2, however $G$ would be replaced by an appropriate subgraph.

**Lemma G.4.** *The following holds*

$$\mathbb{E}[R'] \geq \frac{\epsilon T}{2} - \frac{\epsilon T}{|\mathcal{I}||R|}\sum_{i \in \mathcal{I}}\sum_{j \in [|R|]}\mathrm{d}_{TV}^{\mathcal{F}}\left(\mathcal{Q}_0^i, \mathcal{Q}_j^i\right) + \frac{1}{|\mathcal{I}|}\sum_{i \in \mathcal{I}}\mathbb{E}_i\left[M_i + \frac{N_i}{7}\right]$$

*Proof.* Let $\mathbb{E}_i$ denote the conditional distribution for sampling the active vertex set indexed by $i \in \mathcal{I}$. We have $\mathbb{E}[R'] = \frac{1}{|\mathcal{I}|}\sum_{i \in \mathcal{I}}\mathbb{E}_i[R']$. First let us consider the amount of regret the player incurs for picking a revealing action $N_i$ times. To do this we consider the number of times $1/2 + W_t > 5/6$. The expected number of times this occurs is

$$\mathbb{E}\sum_{t=1}^{T}\mathbb{1}_{1/2+W_t>5/6} \leq \sum_{t=1}^{T}\mathbb{P}\left(|W_t| + \frac{1}{2} \geq \frac{5}{6}\right) \leq \sum_{t=1}^{T}e^{-\frac{1}{d(\rho)\sigma^2}} \leq \sum_{t=1}^{T}e^{-\frac{9\log(T)}{2}} \leq 1.$$

Thus in expectation the regret for picking a revealing action $N_i$ times is at least $(1/6 + \epsilon)(N_i - 1)$. Let $\chi_i$ denote the uniform random variable over $R$ which picks the best action. Denote by $B_j^i$ the number of times action $j$ was played from the active vertices. Then $\mathbb{E}_i[R'] \geq \mathbb{E}_i[\epsilon(T - N_i - B_{\chi_i}^i) + M_i + N_i/6 - 1/6]$. Thus we have

$$\mathbb{E}[R'] = \frac{\sum_{i \in |\mathcal{I}|}\mathbb{E}_i[\epsilon(T - N_i - B_{\chi_i}^i) + M_i + N_i/6 - 1/6]}{|\mathcal{I}|}$$

$$= \epsilon T - \frac{\epsilon}{|\mathcal{I}|}\sum_{i \in \mathcal{I}}\mathbb{E}_i[B_{\chi_i}^i] + \frac{1}{|\mathcal{I}|}\sum_{i \in \mathcal{I}}\mathbb{E}_i\left[M_i + \frac{N_i}{6} - 1/6 - \epsilon N_i\right].$$

Consider $\mathbb{E}_i[B^i_{\chi_i}] = \frac{1}{|R|} \sum_{j \in [|R|]} \mathbb{E}_{\mathcal{Q}^i_j}[B^i_j]$. For each term of the sum we have

$$\mathbb{E}_{\mathcal{Q}^i_j}[B^i_j] - \mathbb{E}_{\mathcal{Q}^i_0}[B^i_j] = \sum_{t=1}^{T}(\mathcal{Q}^i_j(a_t = j) - \mathcal{Q}_0(a_t = j)) \leq T\mathrm{d}^{\mathcal{F}}_{\mathrm{TV}}\left(\mathcal{Q}^i_0, \mathcal{Q}^i_j\right).$$

Thus we get

$$\sum_{i \in \mathcal{I}} \mathbb{E}_i[B^i_{\chi_i}] \leq T\frac{1}{|R|} \sum_{i \in \mathcal{I}} \sum_{j \in [|R|]} \mathrm{d}^{\mathcal{F}}_{\mathrm{TV}}\left(\mathcal{Q}^i_0, \mathcal{Q}^i_j\right) + \frac{1}{|R|} \sum_{i \in \mathcal{I}} \sum_{j \in [|R|]} \mathbb{E}_{\mathcal{Q}^i_0}[B^i_j]$$

$$\leq \frac{T}{|R|} \sum_{i \in \mathcal{I}} \sum_{j \in [|R|]} \mathrm{d}^{\mathcal{F}}_{\mathrm{TV}}\left(\mathcal{Q}^i_0, \mathcal{Q}^i_j\right) + T - \frac{1}{|R|} \sum_{i \in \mathcal{I}} \mathbb{E}_{\mathcal{Q}^i_0}[N_i].$$

Using the assumption that $|\mathcal{I}| \geq 2$, the above implies

$$\mathbb{E}[R'] \geq \frac{\epsilon T}{2} - \frac{\epsilon T}{|\mathcal{I}||R|} \sum_{i \in \mathcal{I}} \sum_{j \in [|R|]} \mathrm{d}^{\mathcal{F}}_{\mathrm{TV}}\left(\mathcal{Q}^i_0, \mathcal{Q}^i_j\right) + \frac{1}{|\mathcal{I}|} \sum_{i \in \mathcal{I}} \mathbb{E}_i\left[M_i + \frac{N_i}{6} - 1/6 - \epsilon N_i\right]$$

Since $\epsilon = \tilde{\Theta}(T^{-1/3})$ we have $\mathbb{E}_i\left[M_i + \frac{N_i - 1}{6} - \epsilon N_i\right] \geq \mathbb{E}_i\left[M_i + \frac{N_i}{7}\right]$. $\qquad\square$

Let $M$ denote the random variable counting the total number of switches and $N$ the random variable denoting the total number of times a revealing action with degree at least 2 was played. We can write $\frac{1}{|\mathcal{I}|} \sum_{i \in \mathcal{I}} \mathbb{E}_i[M_i] \leq \frac{1}{|\mathcal{I}|} \sum_{i \in \mathcal{I}} \mathbb{E}_i[M] = \mathbb{E}[M]$ and similarly $\frac{1}{|\mathcal{I}|} \sum_{i \in \mathcal{I}} \mathbb{E}_i[N_i] \leq \mathbb{E}[N]$. The proof of Theorem 5.2 can now be completed by following the proof of Theorem H.4. We note that bounding $M_i$ by $M$ is in general tight for disjoint union of star graphs and equality occurs for all strategies which switch only between revealing vertices. For general graphs this upper bound can become very loose and we should exercise caution when constructing an upper bound. In particular we should carefully count how many distributions are covered by a single switch.

# H  Lower Bound for Arbitrary Graphs

In this section we propose a construction leading to a non-tight lower bound for general graphs. We choose to present this construction due to it developing tools which can be useful for a tight generic bound. In particular the way we use Lemma H.1 in the proof of Lemma G.2 can be mimicked for general graphs when coupled with a careful counting argument.

Let $G = (V, E)$ be a feedback graph with vertex set $V$ and edge set $E$. Let $\mathcal{I}$ denote the set of all maximal independent sets $I$ of $G$. For any $I$ we say that $I$ is dominated by $S \subseteq V$ if for every $v \in I$, there exists a neighbor of $v$ in $S$. For any $I$ let $S_I$ be a minimal set of vertices which dominates $I$ and let $\mathcal{S}_I$ be the set of all such $S_I$. Let $\delta(S_I)$ equal the maximum number of neighbors in $I$, which a vertex in $S_I$ can have. Let $\delta(\mathcal{S}_I)$ be the maximum over all $\delta(S_I)$ and let $\phi(G) = \min_{I \in \mathcal{I}} \frac{\delta(\mathcal{S}_I)}{|I|}$. Let $I^*$ be a maximal independent set for which $|S_{I^*}| = \phi(G)$. To construct our adversarial loss sequence we begin by uniformly sampling an action $i$ from $I^*$ and setting it to be the action with smallest loss. Let $\mathcal{Q}_i$ denote the conditional probability measure given the sampled best action was $i$ and let $\mathcal{Q}_0$ be the probability distribution when all of the actions in $I^*$ are equal i.e. there is no best action. Let $W_t$ be the stochastic process as defined in Section F. We set the losses for actions in $I^*$ to be $clip(W_t + 1/2)$ for $v \in I^* \setminus \{i\}$ and the loss of $i$ to be $clip(W_t + 1/2 - \epsilon)$. The loss of all other actions is set to be 1. We let $Y_t$ denote the loss vector of observed losses only on $I^*$. WLOG we can disregard other losses, since they will not let us distinguish between $\mathcal{Q}_i$ and $\mathcal{Q}_0$. We denote by $Y_t(j)$ the loss of action $j \in I^*$ if that loss was observed at time $t$. Let $\mathcal{F}$ be the $\sigma$-field generated by $(Y_t)_{t=1}^T$.

Our intuition behind the definition of $\phi(G)$ and the above construction is the following. First we require that the losses based on the stochastic process $(W_t)_{t=1}^T$ be assigned to vertices in an independent set. Otherwise, there would exist a setting in which the best action would be adjacent to another action with losses generated from $(W_t)_{t=1}^T$ and in this case it is information theoretically possible to obtain $O(\sqrt{T})$ regret by playing the best action or its adjacent action enough times, without switching. For every independent set, once a best action is fixed, from the lower bound in

Figure 4: Example of feedback graphs with different $\phi(G)$.

Section F we know two ways to distinguish it. First we switch between the best action and some other action in the independent set (or more generally switch between actions giving information about the best action and another action in the independent set), or play an action which is adjacent to the best action and another action in the independent set. In the general setting there might be an action which is adjacent to multiple actions in the independent set and not adjacent to the best action. In such cases switching between the best action and said action, reveals information proportional to the degree of said action. Similarly if there is an action adjacent to the best action and multiple other actions, selecting it again reveals information proportional to its degree. Since we do not want to assume anything about the strategy of the player, it is natural to select an independent set, such that minimum amount of vertices have a common neighbor. Because the size of the independent set also gives freedom to hide information from the player, we would simultaneously like to maximize its size. This suggests that we search for and independent set which minimizes the ratio in the definition of $\phi(G)$. In Figure 4 we give three examples of graphs with different $\phi(G)$. For the first example the independent set $|I^*|$ is the set of all vertices. The set $S_{I^*}$ is also the set of all vertices and $\delta(S_{I^*}) = 1$ thus $\phi(G) = 1/|V|$ and this is exactly equal to $\gamma(G)^{-1}$. For the second example $I^*$ is the set of leafs of the star graph and $S_{I^*}$ is the vertex adjacent to all other vertices. In this case $\delta(S_{I^*}) = |I^*|$ and $\phi(G) = 1$ which again equals the inverse of the dominating number of $G$. Our final example shows that $\phi(G)$ can be arbitrary close to 1 even though $\gamma(G)^{-1} < 1$. In particular $S_{I^*}$ consists of the bottom 4 vertices and this is also the minimum dominating set of $G$. However, there exists a vertex (the first vertex of the bottom four) of arbitrary large degree so that $\frac{\delta(S_{I^*})}{|I^*|}$ can be arbitrary close to 1. The problem with our lower bound construction becomes clear from this example. The player has a strategy in which too much information is revealed by playing the action of arbitrary large degree. To try and fix this problem we could set only one of the vertices adjacent to the action of large degree according to $(W_t)_{t=1}^T$ and the rest of the adjacent actions are set to have loss equal to 1. This construction can fail for general graphs, as it might happen that there exists another action which is adjacent to exactly the four actions whose losses were chosen according to $(W_t)_{t=1}^T$ in the right most graph of Figure 4.

**Lemma H.1.** *Let $M_i$ be the number of times the player's strategy switched between action adjacent only to $i$ and another action not adjacent to $i$ but adjacent to at least one other action in $I^*$. Let $N_i$ be the number of times the player chose to play an action adjacent to $i$ and another action in $I^*$. Then $d_{TV}^{\mathcal{F}}(\mathcal{Q}_0, \mathcal{Q}_i) \leq \frac{\epsilon}{2\sigma}\sqrt{\omega(\rho)\mathbb{E}_{\mathcal{Q}_0}[|I^*|\phi(G)M_i + N_i]}$.*

*Proof.* Using Yao's minimax principle we can assume the player is deterministic and thus their $t$-th action $a_t$ is a deterministic function of $Y_{0:t-1}$. Using the chain rule for relative entropy and by the construction of $W_t$, we have:

$$D_{KL}\left(\mathcal{Q}_0(Y_{0:T})||\mathcal{Q}_i(Y_{0:T})\right) = D_{KL}\left(\mathcal{Q}_0(Y_0)||\mathcal{Q}_i(Y_1)\right) + \sum_{t=1}^{T} D_{KL}\left(\mathcal{Q}_0(Y_t|Y_{\rho*(t)})||\mathcal{Q}_i(Y_t|Y_{\rho*(t)})\right).$$

Let us consider the term $D_{KL}\left(\mathcal{Q}_0(Y_t|Y_{\rho*(t)})||\mathcal{Q}_i(Y_t|Y_{\rho*(t)})\right)$. First assume that $a_t = a_{\rho(t)}$ is not an action adjacent to $i$ or $a_t = a_{\rho(t)} = i$. Then for any observed $j \in I^*$ we have $Y_t(j) = \mathcal{N}(Y_{\rho(t)}, \sigma^2)$ under both $\mathcal{Q}_0$ and $\mathcal{Q}_i$. Next consider the case when $a_t = a_{\rho(t)}$ is an action adjacent to $i$ and some other $j \in I^*$. In this case $Y_t(j) = Y_t(i) = \mathcal{N}(Y_{\rho(t)}(j), \sigma^2)$ under $\mathcal{Q}_0$ and $Y_t(i) = \mathcal{N}(Y_{\rho(t)}(j) - \epsilon, \sigma^2)$, $Y_t(j) = \mathcal{N}(Y_{\rho(t)}(j), \sigma^2)$ under $\mathcal{Q}_i$ for all observed $j \in I^* \setminus \{i\}$. If $a_t \neq a_{\rho(t)}$ we have 6 options:

1. $a_{\rho(t)}$ is an action adjacent to $i$ and another action $j \in I^* \setminus \{i\}$

(a) $a_t$ is an action adjacent to $i$, in this case $Y_t(j) = Y_t(i) = \mathcal{N}(Y_{\rho(t)}(j'), \sigma^2)$ under $\mathcal{Q}_0$ for all observed $j' \in I^*$ and $Y_t(i) = \mathcal{N}(Y_{\rho(t)}(j) - \epsilon, \sigma^2)$, $Y_t(j') = \mathcal{N}(Y_{\rho(t)}(j), \sigma^2)$ under $\mathcal{Q}_i$ for all observed $j' \in I^*$;

(b) $a_t$ is an action not adjacent to $i$ in this case $Y_t(j') = \mathcal{N}(Y_{\rho(t)}(j), \sigma^2)$ under $\mathcal{Q}_0$ and $Y_t(j') = \mathcal{N}(Y_{\rho(t)}(j), \sigma^2)$ under $\mathcal{Q}_i$ for all observed $j'$ in $I^*$;

2. $a_{\rho(t)}$ is an action not adjacent to $i$ but adjacent to $j$

(a) $a_t$ is an action adjacent to $i$, in this case $Y_t(j') = Y_t(i) = \mathcal{N}(Y_t(j), \sigma^2)$ under $\mathcal{Q}_0$ and $Y_t(i) = \mathcal{N}(Y_{\rho(t)}(j) - \epsilon, \sigma^2)$, $Y_t(j') = \mathcal{N}(Y_{\rho(t)}(j), \sigma^2)$ under $\mathcal{Q}_i$ for all observed $j'$;

(b) $a_t$ is an action not adjacent to $i$, in this case $Y_t(j') = \mathcal{N}(Y_{\rho(t)}(j), \sigma^2)$ under $\mathcal{Q}_0$ and $Y_t(j') = \mathcal{N}(Y_{\rho(t)}(j), \sigma^2)$ under $\mathcal{Q}_i$ for all observed $j'$;

3. $a_{\rho(t)}$ is an action only adjacent to $i$ and no other $j \in I^*$

(a) $a_t$ is an action adjacent to $i$, in this case $Y_t(j') = Y_t(i) = \mathcal{N}(Y_{\rho(t)}(i), \sigma^2)$ under $\mathcal{Q}_0$ and $Y_t(i) = \mathcal{N}(Y_{\rho(t)}(i), \sigma^2)$, $Y_t(j') = \mathcal{N}(Y_{\rho(t)}(j') + \epsilon, \sigma^2)$ under $\mathcal{Q}_i$ for all observed $j'$;

(b) $a_t$ is an action not adjacent to $i$, in this case $Y_t(j') = \mathcal{N}(Y_{\rho(t)}(i), \sigma^2)$ under $\mathcal{Q}_0$ and $Y_t(j') = \mathcal{N}(Y_{\rho(t)}(i) + \epsilon, \sigma^2)$ under $\mathcal{Q}_i$ for all observed $j'$.

Thus we have

$$D_{\text{KL}}\left(\mathcal{Q}_0(Y_t|Y_{\rho*(t)})||\mathcal{Q}_i(Y_t|Y_{\rho*(t)})\right) \leq \frac{\epsilon^2}{2\sigma^2}\mathcal{Q}_0(A_t) + |I^*|\phi(G)\frac{\epsilon^2}{2\sigma^2}\mathcal{Q}_i(B_t)$$

where $A_t$ is the event that $a_{\rho(t)}$ was adjacent to at least one action in $I^* \setminus \{i\}$ and at time $t$ action $i$ was observed and $B_t$ is the event that $a_{\rho(t)}$ was adjacent only to $i$ and the player switched at time $t$ to an action which is adjacent to an action in $I^* \setminus \{i\}$. Let $N_i$ denote the random number of times an action adjacent to $i$ was played and let $M_i$ denote the random number of switches between an action adjacent to $i$ and an action not adjacent to $i$. Let $S_{1:M}$ denote the random sequence of times during which there was a switch. Then we have

$$\sum_{t=1}^{T} \mathbb{1}_{A_t} + \mathbb{1}_{B_t} \leq \sum_{r=1}^{M} \sum_{t \in \text{cut}(S_r)} \mathbb{1}_{A_t} + N_i \leq \omega(\rho)(M_i + N_i),$$

where $\text{cut}(t)$ and $\omega(\rho)$ are defined in Dekel et al. [2014]. Thus

$$D_{\text{KL}}\left(\mathcal{Q}_0(Y_t|Y_{\rho*(t)})||\mathcal{Q}_i(Y_t|Y_{\rho*(t)})\right) \leq \frac{\epsilon^2 \omega(\rho)}{2\sigma^2}\mathbb{E}_{\mathcal{Q}_0}[|I^*|\phi(G)M_i + N_i].$$

Pinsker's inequality that $d_{\text{TV}}^{\mathcal{F}}(\mathcal{Q}_0, \mathcal{Q}_i) \leq \frac{\epsilon}{2\sigma}\sqrt{\omega(\rho)\mathbb{E}_{\mathcal{Q}_0}[|I^*|\phi(G)M_i + N_i]}$. $\qquad\square$

Let $M$ denote the total number of switches and $N$ the total number of times an action revealing adjacent to at least two vertices in $I^*$ is played.

**Lemma H.2.** *It holds that* $\frac{1}{|I^*|}\sum_{i \in I^*} d_{TV}^{\mathcal{F}}(\mathcal{Q}_0, \mathcal{Q}_i) \leq \frac{\epsilon}{\sigma}\sqrt{\frac{\omega(\rho)\phi(G)}{2}}\sqrt{\mathbb{E}_{\mathcal{Q}_0}[M + N]}.$

*Proof.* From concavity of square root and Lemma H.1 we have

$$\frac{1}{|I^*|}\sum_{i \in I^*} d_{TV}^{\mathcal{F}}(\mathcal{Q}_0, \mathcal{Q}_i) \leq \frac{\epsilon\sqrt{\omega(\rho)}}{2\sigma}\sqrt{\frac{1}{|I^*|}\mathbb{E}_{\mathcal{Q}_0}\left[\sum_{i \in I^*} |I^*|\phi(G)M_i + N_i\right]}.$$

Now $\sum_{i \in I^*} M_i = 2M$ since we count each switch twice, once from $i$ and once to $i$. On the other hand each action which is adjacent to $n$ actions in $I^*$ has been overcounted $n$ times. Since $n \leq |I^*|\phi(G)$ we have $\sum_{i \in I^*} N_i \leq |I^*|\phi(G)N$. $\qquad\square$

**Lemma H.3.** *It holds that*

$$\mathbb{E}[R'] \geq \frac{\epsilon T}{2} - \epsilon T \frac{1}{|I^*|}\sum_{i \in I^*} d_{TV}^{\mathcal{F}}(\mathcal{Q}_0, \mathcal{Q}_i) + \mathbb{E}\left[M + \frac{N}{7}\right].$$

*Proof.* First let us consider the amount of regret the player incurs for picking action adjacent to two actions in $I^*$ N times. To do this we consider the number of times $1/2 + W_t > 5/6$. The expected number of times this occurs is

$$\mathbb{E}\sum_{t=1}^{T} \mathbb{1}_{1/2+W_t>5/6} \leq \sum_{t=1}^{T}\mathbb{P}\left(|W_t| + \frac{1}{2} \geq \frac{5}{6}\right) \leq \sum_{t=1}^{T} e^{-\frac{1}{d(\rho)\sigma^2}} \leq \sum_{t=1}^{T} e^{-\frac{9\log(T)}{2}} \leq 1.$$

Thus in expectation the regret for picking an action adjacent to actions in $I^*$ N times is at least $(1/6+\epsilon)(N-1)$. Let $\chi$ denote the uniform random variable over actions in $I^*$, which picks the best action in the beginning of the game. Denote by $B_i$ the number of times action $i \in I^*$ was played. Then $\mathbb{E}[R'] \geq \mathbb{E}[\epsilon(T-N-B_\chi) + M + N/6]$. Thus we have

$$\mathbb{E}[R'] = \frac{\sum_{i\in I^*}\mathbb{E}[\epsilon(T-N-B_i) + M + (N-1)/6 | \chi = i]}{|I^*|}$$

$$= \epsilon T - \frac{\epsilon}{|I^*|}\sum_{i\in I^*}\mathbb{E}_{\mathcal{Q}_i}[B_i] + \mathbb{E}\left[M + \frac{N-1}{6} - \epsilon N\right].$$

Consider $\mathbb{E}_{\mathcal{Q}_i}[B_i]$, we have

$$\mathbb{E}_{\mathcal{Q}_i}[B_i] - \mathbb{E}_{\mathcal{Q}_0}[B_i] = \sum_{t=1}^{T}(\mathcal{Q}_i(a_t=i) - \mathcal{Q}_0(a_t=i)) \leq T\mathrm{d}_{\mathrm{TV}}^{\mathcal{F}}(\mathcal{Q}_0, \mathcal{Q}_i).$$

Thus we get

$$\sum_{i\in I^*}\mathbb{E}_{\mathcal{Q}_i}[B_i] \leq T\sum_{i\in I^*}\mathrm{d}_{\mathrm{TV}}^{\mathcal{F}}(\mathcal{Q}_0, \mathcal{Q}_i) + \sum_{i\in I^*}\mathbb{E}_{\mathcal{Q}_0}[B_i]$$

$$\leq T\sum_{i\in I^*}\mathrm{d}_{\mathrm{TV}}^{\mathcal{F}}(\mathcal{Q}_0, \mathcal{Q}_i) + T - \mathbb{E}_{\mathcal{Q}_0}[N].$$

Using the assumption that $|I^*| \geq 2$, the above implies

$$\mathbb{E}[R'] \geq \frac{\epsilon T}{2} - \frac{\epsilon T}{|I^*|}\sum_{i\in I^*}\mathrm{d}_{\mathrm{TV}}^{\mathcal{F}}(\mathcal{Q}_0, \mathcal{Q}_i) + \mathbb{E}\left[M + \frac{N-1}{6} - \epsilon N\right] + \frac{\epsilon}{2}\mathbb{E}_{\mathcal{Q}_0}[N].$$

Since $\epsilon = \tilde{\Theta}(T^{-1/3})$ we have $\mathbb{E}\left[M + \frac{N-1}{6} - \epsilon N\right] + \frac{\epsilon}{2}\mathbb{E}_{\mathcal{Q}_0}[N] \geq \mathbb{E}\left[M + \frac{N}{7}\right]$ □

**Theorem H.4.** *The expected regret of a deterministic player is at least*

$$\mathbb{E}[R] \geq 4\frac{T^{2/3}}{\log(T)\phi(G)^{1/3}}$$

*Proof.* First assume that the event $M + N/7 > \epsilon T$ does not occur on losses generated from $\mathcal{Q}_0$ or $\mathcal{Q}_i$ for a deterministic player strategy. This implies $\mathcal{Q}_0(M + N/7 > \epsilon T) = \mathcal{Q}_i(M + N/7 > \epsilon T) = 0$. Then

$$\mathbb{E}_{\mathcal{Q}_0}[M + N/7] - \mathbb{E}[M + N/7] = \frac{1}{|I^*|}\sum_{i\in I^*}(\mathbb{E}_{\mathcal{Q}_0}[M + N/7] - \mathbb{E}_{\mathcal{Q}_i}[M + N/7])$$

$$\leq \frac{\epsilon T}{|I^*|}\sum_{i\in I^*}\mathrm{d}_{\mathrm{TV}}^{\mathcal{F}}(\mathcal{Q}_0, \mathcal{Q}_i).$$

The above, together with Lemma H.3 implies

$$\mathbb{E}[R'] \geq \frac{\epsilon T}{2} - \frac{2\epsilon T}{|I^*|}\sum_{i\in I^*}\mathrm{d}_{\mathrm{TV}}^{\mathcal{F}}(\mathcal{Q}_0, \mathcal{Q}_i) + \mathbb{E}_{\mathcal{Q}_0}\left[M + \frac{1}{7}N\right].$$

Applying Lemma F.2 now gives

$$\mathbb{E}[R] \geq \frac{\epsilon T}{3} - \frac{2\epsilon T}{|I^*|}\sum_{i\in I^*}\mathrm{d}_{\mathrm{TV}}^{\mathcal{F}}(\mathcal{Q}_0, \mathcal{Q}_i) + \mathbb{E}_{\mathcal{Q}_0}\left[M + \frac{1}{7}N\right].$$

On the other hand we can bound $\frac{1}{|I^*|} \sum_{i \in I^*} d_{TV}^{\mathcal{F}}(\mathcal{Q}_0, \mathcal{Q}_i)$ by Lemma H.2 as

$$\frac{1}{|I^*|} \sum_{i \in I^*} d_{TV}^{\mathcal{F}}(\mathcal{Q}_0, \mathcal{Q}_i) \leq \frac{\epsilon}{\sigma} \sqrt{\frac{\log(T)\,\phi(G)}{2}} \sqrt{\mathbb{E}_{\mathcal{Q}_0}[M+N]}.$$

This implies

$$\mathbb{E}[R] \geq \frac{\epsilon T}{3} - \frac{\sqrt{2}\epsilon^2 T}{\sigma} \sqrt{\phi(G)\log(T)\,\mathbb{E}_{\mathcal{Q}_0}[M+N]} + \mathbb{E}_{\mathcal{Q}_0}\left[M + \frac{1}{7}N\right].$$

Let $x = \sqrt{\mathbb{E}_{\mathcal{Q}_0}[M+N]}$. Then we have

$$\mathbb{E}[R] \geq \frac{\epsilon T}{3} - \frac{\sqrt{2}\epsilon^2 T \sqrt{\log(T)\,\phi(G)}}{\sigma} x + \frac{x^2}{7}.$$

The quadratic $\frac{x^2}{7} - \frac{\epsilon^2 T \sqrt{2\log(T)\phi(G)}}{\sigma} x$ has global minimum $-\frac{\epsilon^4 T^2 \log(T)\phi(G)}{14}$ We set $\epsilon = c\frac{1}{T^{1/3}\log(T)}$ for a constant $c$ to be determined later. We then have

$$\mathbb{E}[R] \geq \frac{cT^{2/3}}{3\log(T)} - \frac{c^4}{14} \frac{T^{2/3}\phi(G)}{\log(T)^3 \sigma^2}.$$

Set $\sigma = \frac{1}{\log(T)}$. The above implies

$$\mathbb{E}[R] \geq \frac{T^{2/3}}{\log(T)} \left(\frac{c}{3} - \frac{c^4\phi(G)}{14}\right).$$

Choosing $c = \left(\frac{7}{6\phi(G)}\right)^{1/3}$ guarantees $\mathbb{E}[R] \geq \frac{T^{2/3}}{16\log(T)\phi(G)^{1/3}}$.

The case when $M + N/7 > \epsilon T$ is treated in the same way as in the proof of Theorem 5.1 $\qquad\square$

## I  Lower bound for a sequence of feedback graphs in the uninformed setting.

As we already mentioned, the statement of Theorem 1 of Rangi and Franceschetti [2019] does not hold, at least in the informed setting for a fixed feedback graph sequence, where $G_t = G, \forall t \in [T]$. We will show that in the uninformed setting, when we allow the graphs to be chosen by the adversary, there exists a sequence $(G_t)_{t=1}^T$ such that for all $t \in [T]$, $\gamma(G_t) = 1$, $\alpha(G_t) \gg 1$ and $\alpha(G_{1:t}) = \Theta(\alpha(G_t))$, for which any player's strategy will incur regret of the order $\tilde{\Omega}(\alpha(G_{1:t})^{1/3}T^{2/3})$. In particular, there is a non-trivial example of a sequence of graphs for which the independence number is arbitrarily larger than the domination number and every strategy has to incur regret depending on the independence number.

Figure 5: $G_t$

We now present our construction. Fix $\alpha \gg 1$ and let $|V| = 2\alpha$. Let $I$ be a subset of $V$ of size $\alpha$ and let $R = V \setminus I$. Set the losses of actions in $I$ according to the construction of Dekel et al. [2014], as described in Section G. Set the losses of actions in $R$ equal to one. The edges of the graph $G_t = (V, E_t)$ at round $t$ are defined as follows. The vertices in $R$ form a clique. A vertex $r$ is sampled uniformly at random from $R$ to be the revealing action and all edges $(r, v_i), v_i \in I$ are also added to $E_t$. We note that $\alpha(G_t) = \alpha + 1$, $\gamma(G_t) = 1$ for all $t \in [T]$ and $\alpha(G_{1:T}) = \alpha$. We present an illustration for our construction in Figure 5. Here $\alpha = 6$, the set $I$ are the vertices in red, the set $R$ are the vertices in blue.

The intuition behind our construction is that the player needs on average $\alpha$ rounds to observe the losses of all actions, due to the randomization over the revealing vertex $r$. The switching cost again contributes to the $T^{2/3}$ time-horizon regret.

Again assume that the strategy of the player is deterministic. As in Section H, we let $\mathcal{Q}_i$ denote the conditional distribution generated by the observed losses, when the best action was sampled to be $v_i \in I$ and $\mathcal{Q}_0$ denotes the distribution over observed losses when there is no best action in $I$. Let $M_i$ be the number of times the player's strategy switched between an action in $I \setminus \{i\}$ and $i$. Let $M_i'$ be the number of times that the player switched between $i$ and the revealing action. Let $N$ be the total number of times a vertex in $R$ was played and let $N'$ be the total number of times a revealing vertex was played. We have the following.

**Lemma I.1.** *For all* $i \in [|I|] \bigcup \{0\}$

$$\frac{1}{\alpha} \mathbb{E}_{\mathcal{Q}_i}[N] = \mathbb{E}_{\mathcal{Q}_i}[N'].$$

*Proof.* Let $r_t$ denote the revealing action at time $t$.

$$\mathbb{E}_{\mathcal{Q}_i}[N'] = \sum_{t=1}^{T} \mathbb{E}_{\mathcal{Q}_i}[\mathbb{I}(a_t = r_t)] = \sum_{t=1}^{T} \mathcal{Q}_i(a_t \in R) \mathbb{E}_{\mathcal{Q}_i}[\mathbb{I}(a_t = r_t)|a_t \in R]$$

$$+ \sum_{t=1}^{T} \mathcal{Q}_i(a_t \notin R) \mathbb{E}_{\mathcal{Q}_i}[\mathbb{I}(a_t = r_t)|a_t \notin R]$$

$$= \sum_{t=1}^{T} \mathcal{Q}_i(a_t \in R) \mathbb{E}_{\mathcal{Q}_i}[\mathbb{I}(a_t = r_t)|a_t \in R]$$

$$= \sum_{t=1}^{T} \mathcal{Q}_i(a_t \in R) \frac{1}{\alpha} = \frac{1}{\alpha} \sum_{t=1}^{T} \mathbb{E}_{\mathcal{Q}_i}[\mathbb{I}(a_t \in R)] = \frac{1}{\alpha} \mathbb{E}_{\mathcal{Q}_i}[N].$$

This completes the proof. $\qquad\square$

Let $M$ denote the random variable counting the total number of switches.

**Lemma I.2.** *The following inequality holds:* $\frac{1}{\alpha} \sum_{v_i \in I} d_{TV}^{\mathcal{F}}(\mathcal{Q}_0, \mathcal{Q}_i) \leq \frac{\epsilon}{\sigma} \sqrt{\frac{\omega(\rho)}{2\alpha}} \sqrt{\mathbb{E}_{\mathcal{Q}_0}[M + N]}.$

*Proof.* The proof of Lemma H.1 implies that for any $\mathcal{Q}_i$ we have

$$d_{TV}^{\mathcal{F}}(\mathcal{Q}_0, \mathcal{Q}_i) \leq \frac{\epsilon}{2\sigma} \sqrt{\omega(\rho) \mathbb{E}_{\mathcal{Q}_0}[\alpha M_i' + M_i + N']},$$

since the amount of information that can be revealed by a switch is at most $\alpha$ and this precisely happens when the player switches from $i$ to the revealing action. Notice that $\sum_{v_i \in I} M_i' \leq N'$, because the number of switches between any $i$ and a revealing action is bounded by the number of times a revealing action is played. Lemma I.1 implies that $\mathbb{E}_{\mathcal{Q}_0}[\alpha M_i' + M_i + N'] \leq \mathbb{E}_{\mathcal{Q}_0}[N/\alpha + M_i + \alpha M_i']$. Next, we note that $\sum_{i \in [|I|]} M_i \leq 2M$ as each switch is counted at most twice by $M_i$. Thus we have

$$\frac{1}{\alpha} \sum_{v_i \in I} d_{TV}^{\mathcal{F}}(\mathcal{Q}_0, \mathcal{Q}_i) \leq \frac{1}{\alpha} \frac{\epsilon}{2\sigma} \sum_{v_i \in I} \sqrt{\omega(\rho) \mathbb{E}_{\mathcal{Q}_0}[N/\alpha + M_i + \alpha M_i']}$$

$$\leq \frac{\epsilon}{2\sigma} \sqrt{\frac{\omega(\rho)}{\alpha} \mathbb{E}_{\mathcal{Q}_0} \left[ \sum_{v_i \in I} N/\alpha + M_i + \alpha M_i' \right]}$$

$$\leq \frac{\epsilon}{\sigma} \sqrt{\frac{\omega(\rho)}{2\alpha}} \sqrt{\mathbb{E}_{\mathcal{Q}_0}[M + N]},$$

where the second to last inequality follows again from Lemma I.1. $\qquad\square$

Repeating the rest of the arguments in Section H with $\phi(G)$ replaced by $\frac{1}{\alpha}$ shows the following theorem.

**Theorem I.3.** *For any* $\alpha > 1, \alpha \in \mathbb{N}$, *there exists an adversarially generated sequence of feedback graphs* $(G_t)_{t=1}^{T}$, *with* $\alpha(G_t) = \alpha + 1, \gamma(G_t) = 1, \forall t \in [T]$ *and* $\alpha(G_{1:T}) = \alpha$, *such that the expected regret of any strategy in the uninformed setting is at least*

$$\mathbb{E}[R] \geq \frac{\alpha^{1/3} T^{2/3}}{16 \log(T)}.$$