[Reviews · NeurIPS 2019]

Reviewer 1



This work considers nonstochastic bandits with feedback graphs and switching cost. This is the combination of two settings that were previously studied. The regret of bandit learning with a known static feedback graph over K actions scales with sqrt{alpha*T} (ignoring log(K) factors), where alpha is the independent set of the graph, and this is provably tight. In the case of bandits with switching costs, the regret is K^{1/3}*T^{2/3}, which is also provably tight. In earlier work [RF2019], an upper bound of alpha^{1/3}*T^{2/3} was proven for the setting studied in this work. This bound holds even when the graph changes with time and is only revealed to the algorithm after each play (the so-called "uninformed" setting). This work fixes a mistake in the lower bound proven in [RF2019] and shows that alpha^{1/3}*T^{2/3} cannot be improved in the uninformed setting when the adversary controls both the loss sequence and the graph sequence. In the informed setting this works shows that the regret can be improved to \sqrt{beta}*T^{2/3}, where beta is the dominating set of the feedback graph. This is obtained by performing a star decomposition of the graph, running a bandit instance on each star, and combining these instances via a mini-batched version of the corraling algorithm of Agarwal et. al. A lower bound of beta^{1/3} T^{2/3} (as usual ignoring log factors) is proven for the class of disjoint union of stars in the informed setting. A further scenario considers policy regret with losses that depend of the past m actions, where m is fixed, and feedback graphs. The feedback model is peculiar: the loss of adjacent actions is only available when the same action was played in the last m steps. In this setting an upper bound of m^{1/3}*sqrt{b}*T^{2/3} is proven again using the minibatched version of the corraling algorithm. The motivation for combining switched regret with feedback graphs is a bit weak. The paper contains some interesting and clever ideas, especially in the design of the lower bounds. The use of the corraling technique feels suboptimal, and leaves indeed a gap between upper and lower bound. The feedback assumption for the policy regret feels quite arbitrary. Why restricting the feedback graph is needed to prove an upper bound on the regret? In summary, this submission provides suboptimal bounds in a weakly motivated setting. On the other hand, the techniques are nontrivial and there are some clever ideas in the proofs. AFTER AUTHOR FEEDBACK: After reading the author feedback, and after going through the other reviewers' comments, I decided to increase my score by 1 in recognition of the technical contributions.

Reviewer 2



This paper studies adversarial multi-armed bandit with feedback graphs and switching costs: after playing an action, the player observes the losses of all neighboring actions and suffers an additional penalty if there was a switch. The algorithm is based on decomposing the graph into star graphs. For each star graph, they apply a local variant of Exp3 with mini-batches and combine them with a mini-batched variant of Corral. The authors provide upper-bound and lower-bound on the regret in terms of the domination number (\beta) of the graph. Strengths and weaknesses: + I liked the simplicity of the solution to divide the problem into star graphs. The domination number introduced seems to be a natural quantity for this problem. +/- To my opinion, the setting seems somewhat contrived combining feedback graphs and switching costs. The application to policy regret with counterfactual however provides a convincing example that the analysis can be useful and inspire future work. +/- The main part of the paper is rather clear and well written. Yet, I found the proofs in the appendices sometimes a bit hard to follow with sequences of unexplained equations. I would suggest to had some details. - There is a gap between the lower bound and the upper-bound (\sqrt(\beta) instead of \beta^{1/3}). In particular, for some graphs, the existing bound with the independence number may be better. This is also true for the results on the adaptive adversary and the counterfactual feedback. Other remarks: - Was the domination number already introduced for feedback graphs without switching costs? If yes, existing results for this problem should be cited. If not, it would be interesting to state what kind of results your analysis would provide without using the mini-batches. - Note that the length of the mini-batches tau_t may be non-integers. This should be clarified to be sure there are no side effects. For instance, what happens if $\tau_t << 1$? I am not sure if the analysis is still valid. - A better (more formal) definition of the independence and the domination numbers should be provided. It took me some time to understand their meaning. - Alg 1 and Thm 3.1: Since only upper-bounds on the pseudo-regret are provided, the exploration parameter gamma seems to be useless, isn't it? The choice gamma=0 seems to be optimal. A remark on high-probability upper-bounds and the role of gamma might be interesting. In particular, do you think your analysis (which is heavily based on expectations) can be extended to high-probability bounds on the regret? - I understand that this does not suit the analysis (which uses the equivalence in expectation btw Alg1 and Alg6) but it seems to be suboptimal (at least in practice) to discard all the feedbacks obtained while playing non-revealing actions. It would be nice to have practical experiments to understand better if we lose something here. It would be also nice to compare it with existing algorithms. Typos: - p2, l86: too many )) - Thm 3.1: A constant 2 in the number of switches is missing. - p13, l457: some notations seem to be undefined (w_t, W_t). - p14, you may add a remark - p15, l458: the number of switches can be upper-bounded by **twice** the number of times the revealing action is played - p16, l514: I did not understand why Thm 3.1 implies the condition of Thm C.5 with alpha=1/2 and not 1. By the way, (rho_t) should be non-decreasing for this condition to hold.

Reviewer 3



The paper studies adversarial bandits with feedback graph and switching cost among arms. Both the feedback graphs and switching cost for bandit learning have been extensively studied, but combining two aspects together seem to be only covered by one recent prior work Rangi and Franceschetti [2019]. The main technical merit of the paper seems to be connecting the regret upper and lower bound with the dominance number of the feedback graph, denoted as \beta(G). Prior studies only connect regret bounds with independence number \alpha(G), which by definition is greater than or equal to the dominance number. This seems to be a useful advancement in the study of feedback graphs with switching costs. However, the technical result given in the paper is not really a technical improvement of the prior result, because the result of Rangi and Franceschetti [2019] show a regret upper bound of $O(\alpha(G)^{1/3} T^{2/3})$, while the current paper only shows $O(\sqrt{\beta(G)} T^{2/3})$ upper bound. Therefore the term $\sqrt{\beta{G}}$ does not beat $\alpha(G)^{1/3}$ technically, and the authors only conjecture that some adaptation of the algorithm could achieve $\beta{G}^{1/3}$ but was not able to realize that. This could be viewed as a technical limitation of the paper. Besides the above limitations, I also few that some other limitations of the paper prevent it to be more impactful. One is on the motivation. It looks to me that combining feedback graphs with switching cost is more from a theoretical curiosity in extending our knowledge on various bandit algorithms rather than from demanding application scenarios. I do not see a compelling argument in the paper on why this is particularly useful in practice. From this aspect, perhaps the paper would be appreciated more in a more theory oriented conference such as COLT or AISTATS. Another is on the technical novelty of the paper. It looks like both the upper bound and lower bound analyses are based on existing techniques, such as mini-batching and corralling algorithm for the upper bound, and multi-scale random walk for the lower bound. I do clearly see the technical novelty of the paper. The third issue is on empirical evaluation. Given the above limitations, I would feel that some empirical evaluations of the proposed algorithm and comparison with prior work would compensate the incremental nature of the work, but unfortunately the authors do not provide any empirical evidence of the superiority of the proposed algorithm. Overall, I feel that the paper has some interesting theoretical advancement in the theoretical study of bandit algorithms, but I have some reservation on its overall impact, and thus gives the current marginal acceptance score.

[Author Response · NeurIPS 2019]

**Main contribution:** We make significant contributions in this work by completely removing the independence number $\alpha(G)$ from known regret bounds and replace it by the domination number $\beta(G)$. We note that there have been other recent works which also aim at replacing other graph-theoretic quantities like clique partition (e.g. Stochastic Bandits with Side Observations on Networks by Buccapatnam et al. and Graph regret bounds for Thompson Sampling and UCB by Lykouris et al.) by the domination number. These works are motivated by real-world social network graphs, where the clique partition number is much larger compared to the domination number. Note that for star-graphs or star-like graphs, the domination number is substantially smaller than the independence number. It is also important to note that we fix an error in the lower bound of Rangi and Franceschetti.

**Main algorithmic novelty:** Let us emphasize that the main algorithmic novelty in our work is the star graph algorithm (Algorithm 1). We note that the adaptive mini-batch idea is novel and that even though the algorithm seems natural it was non-trivial to prove regret guarantees. For example another natural algorithm, which decides to remain at the current arm with some probability (as in Rangi et al.) can be shown not to achieve a bound depending on the domination number. We also note that applying the corralling algorithm was not straightforward.

**Suboptimality of the bound:** Even though our intuition suggests that our upper bound might be suboptimal, this is only an intuition. It might be the case that our lower bounds are actually not tight and that the optimal regret bound is some interpolating quantity between $\sqrt{\beta(G)}$ and $\alpha(G)^{1/3}$. We wish to stress that we tried other, maybe more natural algorithms based on the star-graph algorithm, which did not require corralling. However, our attempts were unsuccessful and it is very likely that the most natural extension of the star-graph algorithm, which just constructs adaptive mini-batches based on the revealing arms can provably not work. The conjectured suboptimality likely stems from the corralling algorithm and either a more careful analysis or an improved corralling algorithm taking into account the additional disjoint structure of arms and the homogeneous nature of the corralled algorithms are the key to improving the bound. Both tasks, however, are very non-trivial and are left as future work.

**Motivation:** Our motivating examples are stated on lines 28-38 and 198-205. Both the stock market example and the investor example (lines 28-38) are important and do actually occur in real world. Another example is the treatment of patients. In most cases it is harmful to switch between different treatments for the same patient and additional information can be recovered from other patients who were treated differently but have the same disease sub-type. This example also motivates the feedback graph structure arising in the policy regret minimizing section. Indeed we expect to see counterfactual feedback only about the same type of treatment administered for longer periods of time and it is less likely that we get to observe counterfactual feedback about switching treatments often. We would like to note that technically the graph does not need to be restricted in Section 4, however, our algorithm only needs to see the described feedback. This is because of the comparator class against which we compete. We can also give other examples both for the policy regret minimizing setting and switching costs setting.

**Reviewer 1:** Please see paragraphs about motivation, main algorithmic novelty and suboptimality of the bound.

**Reviewer 4:** Please see paragraphs about main contributions, main algorithmic novelty, suboptimality of the bound and motivation. We can add experiments in the extended version of the paper. We would like to emphasize that $\sqrt{\beta(G)} \ll \alpha(G)^{1/3}$ for many graphs. Take for example the star graph or union of star graphs, where $\alpha(G)$ is order number of vertices, while $\beta(G)$ is small or constant!

**Reviewer 2:** We will polish the Appendix more. Regarding the improvement of the regret upper bound, please see the paragraph above on the suboptimality of the bound. We thank the reviewer for pointing out the typos. *Regarding domination number in prior work:* We refer the reviewer to Alon et al. 2015, Buccapatnam et al. 2014 and Lykouris et al. 2019 which we will reference. In general, when there are no switching costs there are two settings in which either $\sqrt{\alpha(G)T}$ is optimal or $\beta(G)^{1/3}T^{2/3}$ is optimal, depending on the available feedback. Please see Appendix A for additional discussion. *Regarding mini-batch sizes:* Mini-batch sizes are always integers. We apologize for the confusion and will add a floor function in the definition. This will not change the final result or the proofs for Section 3 since the important term to control for the regret bound only decreases when we take the floor of $\tau_t$. *Regarding exploration parameter, high probability bounds and discarding feedback:* We thank the reviewer for taking the time to carefully read the paper and notice the extra exploration parameter. This parameter is needed to ensure that $\lfloor \tau_t \rfloor \geq 1$, so that the algorithm terminates. We will fix the typo in Algorithm 1 and elaborate more about the role of the parameter. We also agree that this parameter can help when trying to show high probability regret bounds, however, we expect these to be very non-trivial to show. We also agree that discarding feedback seems prohibitive but we expect that there are adversarial sequences in which the feedback does not really contribute much. For example consider the star graph case, where the leaf arms all have the same loss of 1 and the revealing arm is the best arm. In this case we gain almost no information by playing a leaf arm. In practice it might happen that an algorithm which takes into account the additional information performs better in some settings. We will run experiments to try and verify the above claim. *"- p16, l514: I did not understand why Thm 3.1 implies the condition of Thm C.5 with alpha=1/2 and not 1":* The $\rho_t$ in Theorem C.5 is defined differently than the $\rho$ in Theorem 3.1. In particular the second can be thought of as a squared version of the first and hence we get $\alpha = 1/2$. We will fix this discrepancy in the final version of our work.

[Meta-Review · NeurIPS 2019]

The reviewers felt that the setting considered in the paper was a bit poorly motivated, but the overall evaluation ended up being positive since the reviewers appreciated the nice technical tools introduced in the analysis, and also the contribution of fixing an erroneous lower bound in a previous paper. In the final version, the authors should attempt to provide some stronger motivation for the studied setup.